# Bicriteria Approximation Algorithms for the Submodular Cover Problem

**Wenjing Chen, Victoria G. Crawford**
Department of Computer Science & Engineering
Texas A&M University
jj9754@tamu.edu, vcrawford@tamu.edu

## Abstract

In this paper, we consider the optimization problem Submodular Cover (SCP), which is to find a minimum cardinality subset of a finite universe $U$ such that the value of a submodular function $f$ is above an input threshold $\tau$. In particular, we consider several variants of SCP including the general case, the case where $f$ is additionally assumed to be monotone, and finally the case where $f$ is a regularized monotone submodular function. Our most significant contributions are that: (i) We propose a scalable algorithm for monotone SCP that achieves nearly the same approximation guarantees as the standard greedy algorithm in significantly faster time; (ii) We are the first to develop an algorithm for general SCP that achieves a solution arbitrarily close to being feasible; and finally (iii) we are the first to develop algorithms for regularized SCP. Our algorithms are then demonstrated to be effective in an experimental section on data summarization and graph cut, two applications of SCP.

## 1 Introduction

Submodularity captures a diminishing returns property of set functions: Let $f : 2^U \to \mathbb{R}$ be defined over subsets of a universe $U$ of size $n$. Then $f$ is *submodular* if for all $A \subseteq B \subseteq U$ and $x \notin B$, $f(A \cup \{x\}) - f(A) \geq f(B \cup \{x\}) - f(B)$. Examples of submodular set functions include cut functions in graphs [Balkanski et al., 2018], information-theoretic quantities like entropy and mutual information [Iyer et al., 2021], determinantal point processes [Gillenwater et al., 2012], and coverage functions [Bateni et al., 2017]. Submodular set functions arise in many important real-world applications including active learning [Kothawade et al., 2021, 2022], partial label learning [Bao et al., 2022], structured pruning of neural networks [El Halabi et al., 2022], data summarization [Tschiatschek et al., 2014], and client selection in federated learning [Balakrishnan et al., 2022].

While the majority of existing work has focused on developing approximation algorithms to maximize a submodular function subject to some constraint [Nemhauser et al., 1978a, Mirzasoleiman et al., 2015a, Harshaw et al., 2019, Buchbinder et al., 2014], in this paper we focus on developing algorithms for the related optimization problem of Submodular Cover (SCP), defined as follows.

**Problem 1** (Submodular Cover (SCP)). *Let $f : 2^U \to \mathbb{R}_{\geq 0}$ be a nonnegative submodular set function defined over subsets of the ground set $U$ of size $n$. Given threshold $\tau \leq \max\{f(X) : X \subseteq U\}$, SCP is to find $argmin\{|X| : f(X) \geq \tau\}$.*

SCP captures applications where we seek to achieve a certain value of $f$ in as few elements as possible. For example, consider data summarization, where a submodular function $f$ is formulated to measure how effectively a subset $X$ summarizes the entire dataset $U$ [Tschiatschek et al., 2014]. Then if we set $\tau = \max\{f(X) : X \subseteq U\}$, SCP asks to find the set of minimum size in $U$ that achieves the maximum effectiveness as a summary. Another example is when expected advertising revenue

37th Conference on Neural Information Processing Systems (NeurIPS 2023).

functions are formulated over subsets of a social network [Hartline et al., 2008], then SCP asks how we can reach a certain amount of revenue while picking as small a subset of users as possible.

In this paper, we propose and analyze algorithms for several variants of SCP including the general case, the case where $f$ is assumed to be monotone[1], and finally when $f$ is a regularized monotone submodular function (and potentially takes on negative values). We now list an overview of the contributions of our paper. In addition, a table summarizing all of our algorithmic contributions can be found in Table 1 in the appendix.

(i) We first address the need for scalable algorithms for SCP where $f$ is assumed to be monotone (MSCP). While the greedy algorithm finds the best possible approximation guarantee for MSCP [Feige, 1998], it makes $O(n^2)$ queries of $f$ which may be impractical in many applications. We propose and introduce two algorithms for MSCP which achieve nearly the same theoretical guarantee as the greedy algorithm but only make $O(n \ln(n))$ queries of $f$. In addition, we extend the work of Iyer and Bilmes [2013a] to a method of converting fast randomized approximation algorithms for the dual cardinality constrained monotone submodular maximization problem [Nemhauser et al., 1978b] into approximation algorithms for MSCP.

(ii) Next, we address the need for algorithms that can produce nearly feasible solutions to the general SCP problem, where $f$ can be nonmonotone. In particular, we provide the first algorithm for SCP that returns a solution $S$ that is guaranteed to satisfy: (i) $f(S) \geq (1-\epsilon)\tau$; and (ii) $|S| \leq 2(1+\alpha)|OPT|/\epsilon$ where $OPT$ is an optimal solution to the instance and $\epsilon, \alpha > 0$ are input parameters. A caveat for our algorithm is that it is not necessarily polynomial time and requires an exact solution to the cardinality constrained submodular maximization problem [Buchbinder et al., 2014] on an instance of size $O(|OPT|/\epsilon^2)$.

(iii) Third, we are the first to consider *regularized monotone SCP* (RMSCP). RMSCP is where the objective $f = g - c$ where $g$ is a nonnegative, monotone, and submodular function and $c$ is a modular cost penalty function. $f$ is not necessarily monotone but potentially takes on negative values, and therefore this new problem doesn't fall under the general SCP problem. We develop a method of converting algorithms for the regularized monotone submodular maximization problem [Harshaw et al., 2019] into ones for RMSCP. We then propose the first algorithm for RMSCP, which is a greedy algorithm using queries to a distorted version of $f = g - c$.

(iv) Finally, we conduct an experimental analysis for our algorithms for MSCP and general SCP on instances of data summarization and graph cut. We find that our algorithms for MSCP make a large speedup compared to the standard greedy approach, and we explore the pros and cons of each relative to the other. We also find that our algorithm for general SCP is practical for our instances despite not being guaranteed to run in polynomially many queries of $f$.

## 1.1 Preliminary Definitions and Notation

We first provide a number of preliminary definitions that will be used throughout the paper: (i) The Submodular Maximization Problem (SMP) is the dual optimization problem to SCP defined by, given budget $\kappa$ and nonnegative submodular function $f$, find $\text{argmax}\{f(X) : X \subseteq U, |X| \leq \kappa\}$; (ii) Monotone SCP (MSCP) is the special case of SCP where $f$ is additionally assumed to be monotone; (iii) Regularized MSCP (RMSCP) is where $f = g - c$ and $g$ is monotone, submodular, and nonnegative, while $c$ is a modular[2] nonnegative cost function; (iv) $OPT$ is used to refer to the optimal solution to the instance of SCP that should be clear from the context; (v) $OPT_{SM}$ is used to refer to the optimal solution to the instance of SMP that should be clear from the context; (vi) An $(\alpha, \beta)$-bicriteria approximation algorithm for SCP returns a solution $X$ such that $|X| \leq \alpha|OPT|$ and $f(X) \geq \beta\tau$. An $(\alpha, \beta)$-bicriteria approximation algorithm for SMP returns a solution $X$ such that $f(X) \geq \alpha f(OPT)$ and $|X| \leq \beta\kappa$. Notice that in the $(\alpha, \beta)$ notation, the approximation on the objective is first, and the approximation on the constraint is second; (vii) The marginal gain of adding an element $u \in U$ to a set $S \subseteq U$ is denoted as $\Delta f(S, u) = f(S \cup u) - f(S)$; (viii) The function $f_\tau = \min\{f, \tau\}$.

---

[1] A set function $f$ is monotone if for all $A \subseteq B \subseteq U$, $f(A) \leq f(B)$.
[2] Every $x \in U$ is assigned a cost $c_x$ such that $c(X) = \sum_{x \in X} c_x$.

## 1.2 Related Work

MSCP is the most studied variant of SCP [Wolsey, 1982, Wan et al., 2010, Mirzasoleiman et al., 2015b, 2016, Crawford et al., 2019]. The standard greedy algorithm produces a logarithmic approximation guarantee for MSCP in $O(n^2)$ queries of $f$ [Wolsey, 1982], and this is the best approximation guarantee that we can expect unless NP has $n^{\mathcal{O}(\log(\log(n)))}$-time deterministic algorithms [Feige, 1998]. One version of the greedy algorithm for MSCP works as follows: A set $S$ is initialized to be $\emptyset$. Iteratively, the element $\arg\max\{\Delta f(S, x) : x \in U\}$ is added to $S$ until $f(S)$ reaches $(1 - \epsilon)\tau$. It has previously been shown that this is a $(\ln(1/\epsilon), 1 - \epsilon)$-bicriteria approximation algorithm [Krause et al., 2008]. Beyond greedy algorithms, algorithms for the distributed setting [Mirzasoleiman et al., 2015c, 2016], the streaming setting [Norouzi-Fard et al., 2016], as well as the low-adaptivity setting Fahrbach et al. [2019] for MSCP have been proposed.

On the other hand, developing algorithms for SCP in full generality is more difficult since the monotonicity of $f$ is not assumed. It is not even obvious how to find a feasible solution. The standard greedy algorithm does not have any non-trivial approximation guarantee for SCP. In fact, to the best of our knowledge, no greedy-like algorithms have been found to be very useful for SCP. Recently, Crawford [2023] considered SCP and proved that it is not possible to develop an algorithm that guarantees $f(X) \geq \tau/2$ for SCP in polynomially many queries of $f$ assuming the value oracle model. On the other hand, algorithmic techniques that are used for SMP in the streaming setting [Alaluf et al., 2022] proved to be useful for SCP. In particular, Crawford [2023] proposed an algorithm using related techniques to that of Alaluf et al. that achieves a $(O(1/\epsilon^2), 1/2 - \epsilon)$-bicriteria approximation guarantee for SCP in polynomially many queries of $f$. We also take an approach inspired by the streaming algorithm of Alaluf et al., but sacrifice efficiency in order to find a solution for SCP that is arbitrarily close to being feasible.

SMP has received relatively more attention than SCP [Nemhauser et al., 1978b, Badanidiyuru and Vondrák, 2014, Mirzasoleiman et al., 2015a, Feige et al., 2011, Buchbinder et al., 2014, Alaluf et al., 2022]. Iyer and Bilmes [2013b] proposed a method of converting algorithms for SMP to ones for SCP. In particular, given a deterministic $(\gamma, \beta)$-bicriteria approximation algorithm for SMP, the algorithm `convert` (see pseudocode in Section 4.1 in the supplementary material) proposed by Iyer and Bilmes produces a deterministic $((1 + \alpha)\beta, \gamma)$-bicriteria approximation algorithm for SCP. The algorithm works by making $\log_{1+\alpha}(n)$ guesses for $|OPT|$ (which is unknown in SCP), running the SMP algorithm with the budget set to each guess, and returning the smallest solution with $f$ value above $\gamma\tau$. However, this approach is limited by the approximation guarantees of existing algorithms for SMP. The best $\gamma$ for monotone SMP is $1 - 1/e$, and the best for general SMP where $f$ is not assumed to be monotone is significantly lower [Gharan and Vondrák, 2011]. Several of the algorithms that we propose in this paper do generally follow the model of `convert` in that they rely on guesses of $|OPT|$, but are different because they: (i) Implicitly use bicriteria approximation algorithms for SMP which have better guarantees on the objective ($\gamma$) because they do not necessarily return a feasible solution; (ii) Are more efficient with respect to the number of queries of $f$, since `convert` potentially wastes many queries of $f$ by doing essentially the same behavior for different guesses of $|OPT|$.

## 2 Algorithms and theoretical guarantees

In this section, we present and theoretically analyze our algorithms, with each subsection corresponding to the variant of SCP we consider. In particular, in Section 2.1 we first consider algorithms for MSCP, followed by the general problem of SCP in Section 2.2, and finally in Section 2.3, we consider algorithms for RMSCP.

### 2.1 Monotone submodular cover

In this section, we develop and analyze approximation algorithms for MSCP. The greedy algorithm is a tight $(\ln(1/\epsilon), 1 - \epsilon)$-bicriteria approximation algorithm for MSCP [Krause et al., 2008]. However, the greedy algorithm makes $O(n^2)$ queries of $f$, which is impractical in many application settings with large $U$ and/or when queries of $f$ are costly [Mirzasoleiman et al., 2015a]. Motivated by this, we propose and analyze the algorithms `thresh-greedy-c` and `stoch-greedy-c` for MSCP which give about the same bicriteria approximation guarantees but in many fewer queries of $f$.

We first describe `thresh-greedy-c`. `thresh-greedy-c` is closely related to the existing threshold greedy algorithm for monotone SMP (MSMP) [Badanidiyuru and Vondrák, 2014], and therefore we relegate the pseudocode of `thresh-greedy-c` to Section 4.1 in the supplementary material and only include a brief discussion here. At each iteration of `thresh-greedy-c`, instead of picking the element with the highest marginal gain into $S$, it sequentially adds any elements in $U$ with marginal gain above a threshold, $w$, at the time of addition. $w$ is initialized to $\max_{u \in U} f(\{u\})$, and is decreased by a factor of $(1 - \epsilon/2)$ when the algorithm proceeds to the next iteration. `thresh-greedy-c` adds elements to a solution $S$ until $f(S)$ reaches $(1 - \epsilon)\tau$, which is shown to happen in at most $\ln(2/\epsilon)|OPT| + 1$ elements in the proof of Theorem 1. We now state the theoretical guarantees of `thresh-greedy-c` in Theorem 1.

**Theorem 1.** *`thresh-greedy-c` produces a solution with $(\ln(2/\epsilon) + 1, 1 - \epsilon)$-bicriteria approximation guarantee to MSCP, in $O(\frac{n}{\epsilon} \log(\frac{n}{\epsilon}))$ number of queries of $f$.*

Another method of speeding up the standard greedy algorithm is by introducing randomization, as has been done for MSMP in the stochastic greedy algorithm [Mirzasoleiman et al., 2015a]. A natural question is whether a randomized algorithm for MSMP can be converted into an algorithm for MSCP using the algorithm `convert` of Iyer and Bilmes [2013b]. However, `convert` relies on a deterministic approximation guarantee. We now introduce a new algorithm called `convert-rand` that is analogous to `convert` but runs the MSMP algorithm repeatedly in order to have the approximation guarantee hold with high probability. Pseudocode for `convert-rand`, as well as a proof of Theorem 2 can be found in Section 4.1 in the supplementary material.

**Theorem 2.** *Any randomized $(\gamma, \beta)$-bicriteria approximation algorithm for MSMP that runs in time $\mathcal{T}(n)$ where $\gamma$ holds only in expectation can be converted into an approximation algorithm for MSCP that with probability at least $1 - \delta$ is a $((1 + \alpha)\beta, \gamma - \epsilon)$-bicriteria approximation algorithm that runs in time $O(\log_{1+\alpha}(|OPT|) \ln(1/\delta)\mathcal{T}(n)/\ln(\frac{1-\gamma+\epsilon}{1-\gamma}))$.*

Therefore by applying Theorem 2 to the stochastic greedy algorithm of Mirzasoleiman et al., we have a $(1 + \alpha, 1 - 1/e - \epsilon)$-bicriteria approximation algorithm for MSCP with high probability in $O(n \log_{1+\alpha}(|OPT|) \ln(1/\delta) \ln(1/\epsilon)/\ln(\frac{1-\gamma+\epsilon}{1-\gamma}))$ queries of $f$. However, a factor of $1 - 1/e - \epsilon$ of $\tau$ is not very close to feasible, and further the `convert-rand` method wastes many queries of $f$ essentially doing the same computations for different guesses of $|OPT|$. Therefore we focus the rest of this section on developing an algorithm, `stoch-greedy-c`, that uses the techniques of the stochastic greedy algorithm more directly for MSCP.

The idea behind the stochastic greedy algorithm for MSMP is that instead of computing the marginal gains of all elements at each iteration, we take a uniformly random sampled subset from $U$ and pick the element with the highest marginal gain among the sampled subset. If the sampled subset is sufficiently large, in particular of size at least $(n/\kappa) \ln(1/\epsilon)$ where $\kappa$ is the budget for the instance of MSMP and $\epsilon > 0$ is an input, then with high probability a uniformly random element of $OPT_{SM}$ will appear in the sampled subset and the marginal gain of adding the element is nearly the same as the standard greedy algorithm in expectation. However, in MSMP we know that $|OPT_{SM}| = \kappa$, but in MSCP $|OPT|$ is unknown. Therefore it is not obvious how to apply this technique in a more direct way than `convert-rand`.

We now introduce our algorithm `stoch-greedy-c` for MSCP, pseudocode for which is provided in Algorithm 1. `stoch-greedy-c` takes as input $\epsilon > 0$, $\delta > 0$, $\alpha > 0$, and an instance of MSCP. `stoch-greedy-c` keeps track of $O(\ln(1/\delta))$ possibly overlapping solutions $S_1, S_2, ...$ throughout a sequence of iterations. `stoch-greedy-c` also keeps track of an estimate of $|OPT|$, $g$. During each iteration, for each solution $S_i$, `stoch-greedy-c` uniformly randomly and independently samples a set $R$ of size $\min\{n, (n/g) \ln(3/\epsilon)\}$ and adds $u = \text{argmax}\{\Delta f_\tau(S_i, x) : x \in R\}$ to $S_i$. Every time $\frac{\alpha}{1+\alpha} \ln(3/\epsilon)g$ elements have been added to each $S_i$, $g$ is increased by a factor of $1 + \alpha$. `stoch-greedy-c` stops once there exists an $S_i$ such that $f(S_i) \geq (1 - \epsilon)\tau$, and returns this solution.

We now state the theoretical results for `stoch-greedy-c` in Theorem 3.

**Theorem 3.** *Suppose that `stoch-greedy-c` is run for an instance of MSCP. Then with probability at least $1 - \delta$, `stoch-greedy-c` outputs a solution $S$ that satisfies a $((1 + \alpha)\lceil \ln(3/\epsilon) \rceil, 1 - \epsilon)$-bicriteria approximation guarantee in at most $O\left(\frac{\alpha}{1+\alpha} n \ln(1/\delta) \ln^2(3/\epsilon) \log_{1+\alpha}(|OPT|)\right)$ queries of $f$.*

---

**Algorithm 1** `stoch-greedy-c`

---

**Input**: $\epsilon, \alpha, \delta$
**Output**: $S \subseteq U$

1:  $S_i \leftarrow \emptyset \; \forall i \in \{1, ..., \ln(1/\delta)/\ln(2)\}$
2:  $r \leftarrow 1, g \leftarrow 1 + \alpha$
3: **while** $f(S_i) < (1 - \epsilon)\tau \; \forall i$ **do**
4:     **for** $i \in \{1, ..., \ln(1/\delta)/\ln(2)\}$ **do**
5:         $R \leftarrow$ sample $\min\{n, n\ln(3/\epsilon)/g\}$ elements from $U$
6:         $u \leftarrow \text{argmax}_{x \in R} \Delta f_\tau(S_i, x)$
7:         $S_i \leftarrow S_i \cup \{u\}$
8:     $r \leftarrow r + 1$
9:     **if** $r > \ln(3/\epsilon)g$ **then** $g \leftarrow (1 + \alpha)g$
10: **return** $\text{argmin}\{|S_i| : f(S_i) \geq (1 - \epsilon)\tau\}$

---

Compared to `thresh-greedy-c`, `stoch-greedy-c` has a better dependence on $\epsilon$ in terms of the number of queries made to $f$. In addition, it is possible to extend the stochastic greedy algorithm of Mirzasoleiman et al. to a $(1 - \epsilon, \ln(1/\epsilon))$-bicriteria approximation algorithm for MSMP and then use `convert-rand` (see Section 4.1 in the supplementary material). However, `stoch-greedy-c` still would have strictly fewer queries of $f$ by a factor of $\frac{\alpha}{1+\alpha}$ compared to this approach because `convert-rand` does essentially the same computations for different guesses of $|OPT|$.

In order to prove Theorem 3, we first need Lemma 1 below, which states that as long as $g \leq (1 + \alpha)|OPT|$, the marginal gain of adding $u$ in Line 6 is about the same as the standard greedy algorithm in expectation. Next, Lemma 2 below uses Lemma 1 to show that by the time $g$ reaches $(1 + \alpha)|OPT|$, $\mathbb{E}[f_\tau(S_i)] \geq (1 - \frac{\epsilon}{2})\tau$ for all $i$. Finally, because there are $O(\ln(1/\delta))$ solutions, by the time $g$ reaches $(1 + \alpha)|OPT|$, there exists $i$ such that $f(S_i) \geq (1 - \epsilon)\tau$ with probability at least $1 - \delta$ by using concentration bounds, which is stated in Lemma 3. Because of Lemma 3 we keep increasing $g$ by a factor of $(1 + \alpha)$ periodically, because intuitively the longer we keep adding elements, the bigger we know that $|OPT|$ must be since the algorithm is still running and none of the solution sets has reached $(1 - \epsilon)\tau$ yet. The proof of Lemmas 1 and 2, and of Theorem 3 can be found in Section 4.1 in the supplementary material.

**Lemma 1.** *Consider any of the sets $S_i$ at the beginning of an iteration on Line 4 where $g \leq (1 + \alpha)|OPT|$. Then if $u_i$ is the random element that will be added on Line 6, we have that* $\mathbb{E}[\Delta f_\tau(S_i, u_i)] \geq \frac{1 - \epsilon/3}{(1+\alpha)|OPT|}(\tau - f_\tau(S_i))$.

**Lemma 2.** *Once $r$ reaches $(1 + \alpha)\lceil \ln(3/\epsilon)|OPT| \rceil$, we have that $\mathbb{E}[f_\tau(S_i)] \geq \left(1 - \frac{\epsilon}{2}\right)\tau$ for all $i$.*

**Lemma 3.** *With probability at least $1 - \delta$, once $r$ reaches $(1 + \alpha)\lceil \ln(3/\epsilon)|OPT| \rceil$, we have that $\max_i f(S_i) \geq (1 - \epsilon)\tau$.*

## 2.2   Non-monotone submodular cover

In this section, we introduce and theoretically analyze the algorithm `stream-c` for SCP in the general setting, where $f$ is not assumed to be monotone. In the general setting, the standard greedy algorithm doesn't have non-trivial approximation guarantee for SCP. In addition, it has previously been shown that it is not possible for an algorithm to guarantee that $f(X) \geq \tau/2$ for SCP, where $X$ is its returned solution, in polynomially many queries of $f$ assuming the value oracle model [Crawford, 2023]. Our algorithm `stream-c` *does* produce a solution $X$ that is guaranteed to satisfy $f(X) \geq (1 - \epsilon)\tau$, but relies on solving an instance of SMP exactly on a set of size $O(|OPT|/\epsilon^2)$. Despite not being polynomial time, `stream-c` is still useful for some instances of SCP because: (i) $|OPT|$ may be relatively small; and (ii) the instance of SMP may be relatively easy to solve, e.g. $f$ may be very close to monotone on the instance of SMP even if it was very non-monotone on the original instance of SCP. These aspects of `stream-c` are further explored in Section 3.

We now describe `stream-c`, pseudocode for which can be found in Algorithm 2. `stream-c` takes as input $\epsilon > 0$, $\alpha > 0$, and an instance of SCP. `stream-c` takes sequential passes through the universe $U$ (Line 4) with each pass corresponding to a new guess of $|OPT|$, $g$. $g$ is initialized as $1 + \alpha$, and at the end of each pass is increased by a factor of $1 + \alpha$. Throughout `stream-c`, a subset

---

**Algorithm 2** `stream-c`

---

**Input**: $\epsilon, \alpha$
**Output**: $S \subseteq U$

1: $S \leftarrow \emptyset, S_1 \leftarrow \emptyset, ..., S_{2/\epsilon} \leftarrow \emptyset$
2: $g \leftarrow 1 + \alpha$
3: **while** $f(S) < (1 - \epsilon)\tau$ **do**
4:     **for** $u \in U$ **do**
5:         **if** $\exists j$ s.t. $\Delta f(S_j, u) \geq \epsilon\tau/(2g)$ and $|S_j| < 2g/\epsilon$ **then**
6:             $S_j \leftarrow S_j \cup \{u\}$
7:     $S \leftarrow \operatorname{argmax}\{f(X) : X \subseteq \cup_{i=1}^{2/\epsilon} S_i, |X| \leq 2g/\epsilon\}$
8:     $g = (1 + \alpha)g$
9: **return** $S$

---

of elements of $U$ are stored into $2/\epsilon$ disjoint sets, $S_1, ..., S_{2/\epsilon}$. An element $u$ is stored in at most one set $S_j$ if both of the following are true: (i) $|S_j| < 2g/\epsilon$; (ii) adding $u$ is sufficiently beneficial to increasing the $f$ value of $S_j$ i.e. $\Delta f(S_j, u) \geq \epsilon\tau/(2g)$. If no such $S_j$ exists, $u$ is discarded. At the end of each pass, `stream-c` finds $S = \operatorname{argmax}\{f(X) : X \subseteq \cup S_i, |X| \leq 2g/\epsilon\}$ on Line 7. If $f(S) \geq (1 - \epsilon)\tau$, then $S$ is returned and `stream-c` terminates. We now present the theoretical guarantees of `stream-c` in Theorem 4.

**Theorem 4.** *Suppose that* `stream-c` *is run for an instance of SCP. Then* `stream-c` *returns* $S$ *such that* $f(S) \geq (1 - \epsilon)\tau$ *and* $|S| \leq (1 + \alpha)(2/\epsilon)|OPT|$ *in at most*

$$\log_{1+\alpha}(|OPT|) \left( \frac{2n}{\epsilon} + \mathcal{T}\left( (1 + \alpha)\left( \frac{4}{\epsilon^2}|OPT| \right) \right) \right)$$

*queries of* $f$, *where* $\mathcal{T}(m)$ *is the number of queries to* $f$ *of the algorithm for SMP used on Line 7 of Algorithm 2 on an input set of size* $m$.

The key idea for proving Theorem 4 is that by the time $g$ is in the region $[|OPT|, (1+\alpha)|OPT|]$, there exists a subset $X \subseteq \cup S_i$ such that $|X| \leq 2g/\epsilon$ and $f(X) \geq (1-\epsilon)\tau$. In fact, it is shown in the proof of Lemma 4 in Section 4.2 of the supplementary material that the set $X$ is $S_t \cup (\cup_i S_i \cap OPT)$ for a certain one of the sets $S_t$. Then when we solve the instance of SMP on Line 7, we find a set that has these same properties as $X$, and `stream-c` returns this set and terminates. Because $g \leq (1 + \alpha)|OPT|$, the properties described in Theorem 4 hold. Further notice that $|\cup_i S_i| \leq 2(1 + \alpha)|OPT|/\epsilon^2$ at all times before `stream-c` exits, which implies the bounded query complexity in Theorem 4. The key idea for proving Theorem 4 is stated below in Lemma 4 and proven in Section 4.2 in the supplementary material.

**Lemma 4.** *By the time that* $g$ *reaches the region* $[|OPT|, (1 + \alpha)|OPT|]$ *and the loop on Line 4 of* `stream-c` *has completed, there exists a set* $X \subseteq \cup S_i$ *of size at most* $2(1 + \alpha)|OPT|/\epsilon$ *such that* $f(X) \geq (1 - \epsilon)\tau$.

### 2.3 Regularized monotone submodular cover

The final class of submodular functions we consider take the form $f = g - c$ where $g$ is monotone, submodular, and nonnegative, while $c$ is a modular, nonnegative penalty cost function, called the Regularized Monotone Submodular Cover Problem (RMSCP). In this case, $f$ may take on negative values and therefore this class of submodular functions does not fit into general SCP. $f$ may also be nonmonotone. Existing theoretical guarantees for the dual problem of Regularized Monotone Submodular Maximization (RMSMP) are in a different form than typical approximation algorithms [Harshaw et al., 2019, Kazemi et al., 2021]. In particular, they are of the following form: Given budget $\kappa$, the RMSMP algorithm is guaranteed to return a set $S$ such that $|S| \leq \kappa$ and $g(S) - c(S) \geq \gamma g(OPT_{SM}) - c(OPT_{SM})$ where $\gamma$ is some value less than 1, e.g. $1 - 1/e$ for the distorted greedy algorithm of Harshaw et al.. A guarantee of this form means `convert` cannot be used (the check on Line 2 of the pseudocode for `convert` in the appendix is the problem). Motivated by this, we first develop an algorithm, `convert-reg`, that takes algorithms for RMSMP and converts them into an algorithm for RMSCP. Next, we propose a generalization of the distorted greedy algorithm of Harshaw et al. for RMSMP, called `distorted-bi`, that can be used along with `convert-reg` to produce an algorithm for RMSCP.

---

**Algorithm 3** `convert-reg`

---

**Input**: $\alpha > 0$
**Output**: $S \subseteq U$

1: $\kappa \leftarrow 1 + \alpha, S \leftarrow \emptyset$
2: **while** $g(S) - \frac{\gamma}{\beta}c(S) < \gamma\tau$ **do**
3:     $S \leftarrow$`reg` run with objective $g - \frac{\gamma}{\beta}c$ and budget $\kappa$
4:     $\kappa \leftarrow (1 + \alpha)\kappa$
5: **return** $S$

---

We now describe `convert-reg`, pseudocode for which can be found in Algorithm 3. `convert-reg` takes as input an algorithm `reg` for RMSMP with the guarantees described previously, and $\alpha > 0$. `convert-reg` repeatedly makes guesses for $|OPT|$, $\kappa$. For each guess $\kappa$, the algorithm `reg` is run on an instance of RMSMP with objective $g - (\gamma/\beta)c$ and budget $\kappa$. Once $g - (\gamma/\beta)c$ reaches $\gamma\tau$, `convert-reg` exits.

The theoretical guarantees of `convert-reg` are stated below in Theorem 5 and proven in Section 4.3 in the supplementary material. Theorem 5 makes a slightly stronger assumption on `reg` than its approximation guarantees relative to $OPT_{SM}$. In particular, it is assumed that it returns a solution satisfying $|S| \leq \rho\kappa$ and $g(S) - c(S) \geq \gamma g(X) - \beta c(X)$ *for all* $X \subseteq U$ *such that* $|X| \leq \kappa$, not just for $OPT_{SM}$. However, this is true of many algorithms for RMSMP including the distorted greedy algorithm of Harshaw et al..

**Theorem 5.** *Suppose that we have an algorithm* `reg` *for RMSMP, and given budget $\kappa$* `reg` *is guaranteed to return a set $S$ of cardinality at most $\rho\kappa$ such that $g(S) - c(S) \geq \gamma g(X) - \beta c(X)$ for all $X$ such that $|X| \leq \kappa$, in time $T(n)$. Then the algorithm* `convert-reg` *using* `reg` *as a subroutine returns a set $S$ in time $O(\log_{1+\alpha}(n)T(n))$ such that $|S| \leq (1 + \alpha)\rho|OPT|$ and $g(S) - \frac{\gamma}{\beta}c(S) \geq \gamma\tau$.*

If we use `convert-reg` on the distorted greedy algorithm of Harshaw et al., we end up with an algorithm for RMSCP that is guaranteed to return a set $S$ such that $|S| \leq (1 + \alpha)|OPT|$ and $g(S) - (1 - 1/e)c(S) \geq (1 - 1/e)\tau$. If we set $c = 0$, then the problem setting reduces to MSCP and the distorted greedy algorithm of Harshaw et al. [2019] is equivalent to the standard greedy algorithm. However, our approximation guarantee does not reduce to the $(\ln(1/\epsilon), 1-\epsilon)$-bicriteria approximation guarantee that would be preferable. A more intuitive result would be one that converges to that of the standard greedy algorithm as $c$ goes to 0. Motivated by this, we now propose an extension of the distorted greedy algorithm of Harshaw et al. [2019] for RMSMP, `distorted-bi`, that accomplishes this.

We now describe `distorted-bi`, pseudocode for which can be found in Section 4.3 in the supplementary material. `distorted-bi` takes as input an instance of RMSMP and $\epsilon > 0$. `distorted-bi` is related to the standard greedy algorithm, but instead of making queries to $g - c$, `distorted-bi` queries a distorted version of $g - c$ that de-emphasizes $g$ compared to $c$, and evolves over time. In particular, when element $i$ is being added to the solution set, we choose the element of maximum marginal gain, provided it is positive, to the objective

$$\Phi_i(X) = \left(1 - \frac{1}{\kappa}\right)^{\ln(1/\epsilon)\kappa - i} g(X) - c(X).$$

The theoretical guarantees of `distorted-bi` are now presented in Theorem 6, and the proof of Theorem 6 can be found in Section 4.3 in the supplementary material.

**Theorem 6.** *Suppose that* `distorted-bi` *is run for an instance of RMSMP. Then* `distorted-bi` *produces a solution $S$ in $O(n\kappa \ln(1/\epsilon))$ queries of $f$ such that $|S| \leq \ln(1/\epsilon)\kappa$ and for all $X \subseteq U$ such that $|X| \leq \kappa$, $g(S) - c(S) \geq (1 - \epsilon)g(X) - \ln(1/\epsilon)c(X)$.*

Therefore by running `convert-reg` with `distorted-bi` as a subroutine for RMSMP, we end up with an algorithm for RMSCP that is guaranteed to return a set $S$ such that $|S| \leq (1 + \alpha)\ln(1/\epsilon)|OPT|$ and $g(S) - (1 - \epsilon)c(S)/\ln(1/\epsilon) \geq (1 - \epsilon)\tau$ in $O((1 + \alpha)n|OPT|\log_{1+\alpha}(|OPT|)\log(1/\epsilon))$ queries of $f$.

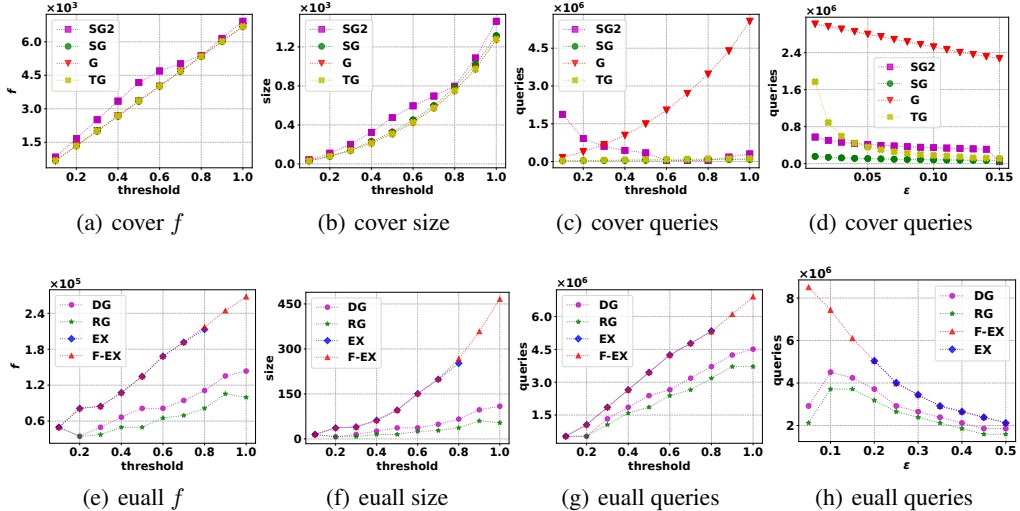

| (a) cover $f$ | (b) cover size | (c) cover queries | (d) cover queries |
|---|---|---|---|

| (e) euall $f$ | (f) euall size | (g) euall queries | (h) euall queries |
|---|---|---|---|

Figure 1: The experimental results of running the monotone algorithms on instances of data summarization on the delicious URL dataset ("cover") and running `stream-c` on the instances of graph cut on the email-EuAll graph ("euall").

## 3 Experiments

In this section, we experimentally evaluate the algorithms proposed in Sections 2.1 and 2.2. In particular, the emphasis of Section 3.1 is on evaluation of our algorithm `stoch-greedy-c` on instances of data summarization, an application of MSCP. Next, we evaluate `stream-c` on instances of graph cut, an application of SCP that is not monotone, in Section 3.2. Additional details about the applications, setup, and results can be found in Section 5 in the supplementary material.

### 3.1 Monotone submodular objective

We first compare the solutions returned by `stoch-greedy-c` ("SG"), `greedy-c` ("G"), `thresh-greedy-c` ("TG"), and `convert-rand` using the *bicriteria* extension of the stochastic greedy algorithm of Mirzasoleiman et al. (see Section 4.1 in the supplementary material) ("SG2") on instances of data summarization. The data summarization instance featured here in the main paper is the delicious dataset of URLs tagged with topics, and $f$ takes a subset of URLs to the number of distinct topics represented by those URLs ($n = 5000$ with 8356 tags) [Soleimani and Miller, 2016]. Additional datasets are explored in Section 5.2 in the supplementary material. We run the algorithms with input $\epsilon$ in the range $(0, 0.15)$ and threshold values between 0 and $f(U)$ ($f(U)$ is the total number of tags). When $\epsilon$ is varied, $\tau$ is fixed at $0.6f(U)$. When $\tau$ is varied, $\epsilon$ is fixed at $0.2$. The parameter $\alpha$ is set to be $0.1$ and the initial guess of $|OPT|$ for `stoch-greedy-c` and `convert-rand` is set to be $\tau/\max_s f(s)$.

The results in terms of the $f$ values and size of the solutions are presented in Figure 1(a) and 1(b). From the plots, one can see that the $f$ values and size of solutions returned by `stoch-greedy-c`, `greedy-c`, `thresh-greedy-c` are nearly the same, and are smaller than the ones returned by `convert-rand`. This is unsurprising, because the theoretical guarantees on $f$ and size are about the same for the different algorithms, but `convert-rand` tends to perform closer to its worst case guarantee on size. The number of queries to $f$ for different $\epsilon$ and $\tau$ are depicted in Figures 1(d) and 1(c). Recall that the theoretical worst case number of queries to $f$ for `stoch-greedy-c`, `greedy-c`, `thresh-greedy-c` and `convert` are $O((\alpha/(1+\alpha))n\ln^2(1/\epsilon)\log_{1+\alpha}(|OPT|))$, $O(n\ln(1/\epsilon)|OPT|)$, $O(n\log(|OPT|/\epsilon)/\epsilon)$, and $O(n\ln^2(1/\epsilon)\log_{1+\alpha}(|OPT|)$ respectively. As expected based on these theoretical guarantees, `greedy-c` does the worst and increases rapidly as $\tau$ (and therefore $|OPT|$) increases. `thresh-greedy-c` tends to do worse compared to `stoch-greedy-c` and `convert` as $\epsilon$ gets smaller. `stoch-greedy-c` consistently performs the fastest out of all of the algorithms.

## 3.2 Non-Monotone Submodular Objective

We now analyze the performance of `stream-c` on several instances of graph cut over real social network data. The universe $U$ is all nodes in the network, and $f$ is the number of edges between a set and its complement. The network featured in the main paper is the email-EuAll dataset ($n = 265214$, 420045 edges) from the SNAP large network collection [Leskovec and Sosič, 2016] and additional datasets can be found in Section 5.1 in the supplementary material. We run `stream-c` with input $\epsilon$ in the range $(0, 0.5)$ and threshold values between 0 and $f(X)$ where $X$ is a solution returned by the unconstrained submodular maximization algorithm of Buchbinder et al. [2015] on the instance. When $\epsilon$ is varied, $\tau$ is fixed at $0.9f(X)$. When $\tau$ is varied, $\epsilon$ is fixed at 0.15.

We compare the performance of `stream-c` using several possible algorithms for the subroutine of SMP over $\cup S_i$ (see line 7 in Algorithm 2), including a polynomial time approximation algorithm and an unconstrained submodular maximization algorithm. In particular, we use the random greedy approximation algorithm for SMP that is proposed in Buchbinder et al. [2014] ("RG"), and the double greedy approximation algorithm for unconstrained submodular maximization proposed in Buchbinder et al. [2015] ("DG"). Random greedy and double greedy are both approximation algorithms ($1/e$ in expectation and $1/2$ in expectation respectively), and therefore the stopping conditions are set to be $\frac{(1-\epsilon)\tau}{e}$ and $\frac{(1-\epsilon)\tau}{2}$ respectively. We also consider an exact algorithm ("EX"), which essentially is a greedy heuristic followed by an exact search of all (exponentially many) possible solutions if the greedy fails. On instances where the exact algorithm was unable to complete in a time period of 5 minutes, we did not include a data point. We further discuss the use of these algorithms in Section 5.1 in the supplementary material.

Before introducing the fourth subroutine, we discuss an interesting pattern that we saw in our instances of graph cut. We noticed that it was often the case that: (i) $\cup S_i$ tended to be small compared to its upper bound and in fact typically $|\cup S_i|$ was smaller than the SMP constraint, making the subroutine an instance of unconstrained submodular maximization; (ii) The majority of elements (if not all) were "monotone" in the sense that for many $x \in \cup S_i$, $\Delta f(\cup S_i/x, x) \geq 0$. Let $M \subseteq \cup S_i$ be the set of monotone elements. It follows that if (i) holds, then the instance of submodular maximization is equivalent to $\arg\max_{X \in \cup S_i/M} f(X \cup M)$. If $M$ is large in $\cup S_i$, this new problem instance is relatively easy to solve exactly. This motivates our fourth algorithm, fast-exact ("F-EX"), used on instances where (i) holds, and is to separate $\cup S_i$ into monotone and non-monotone and search for the best subset amongst the non-monotone elements in a similar manner as the plain exact algorithm. We explore to what extent properties (i) and (ii) hold on different instances, as well as give additional details about the fast exact algorithm, in Section 5.2 in the supplementary material.

The results in terms of the $f$ values and size of the output solutions returned by the four algorithms are plotted in Figure 1(e) and Figure 1(f). From the plots, one can see that the $f$ values satisfy that $f(S_{\text{exact}}) \approx f(S_{\text{f-exact}}) > f(S_{\text{DG}}) > f(S_{\text{RG}})$. This is due to the stopping conditions for each algorithm, which follow from each algorithms approximation guarantee on $f$ of $1 - \epsilon$, $1 - \epsilon$, $1/2$, and $1/e$ respectively. On the other hand, the size mirrors the $f$ value, since it tends to be the case that reaching a higher $f$ value requires more elements from $U$. The number of queries made by the algorithms can be seen in Figure 1(h) and 1(g). As expected, the exact algorithms make more queries compared to the approximation algorithms, and in some cases "EX" doesn't even finish. However, by taking advantage of the properties (i) and (ii) discussed above, "F-EX" is able to run even for smaller $\epsilon$. Therefore, depending on the application, an exact algorithm on the relatively small set $\cup S_i$ may be a practical choice in order to achieve a solution that is very close to feasible.

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
