Table 1: Theoretical guarantees of a subset of algorithms in this paper

| Alg name | $f$ value | soln size | number of queries |
|---|---|---|---|
| `greedy-c` | $(1-\epsilon)\tau$ | $\ln(1/\epsilon)$ | $O(n\ln(1/\epsilon)|OPT|)$ |
| `stoch-greedy-c` | $(1-\epsilon)\tau$ | $(1+\alpha)\ln(3/\epsilon)$ | $O(\frac{\alpha}{1+\alpha}n\ln(1/\delta)\ln^2(1/\epsilon)\log_{1+\alpha}(|OPT|))$ |
| `thresh-greedy-c` | $(1-\epsilon)\tau$ | $\ln(2/\epsilon)+1$ | $O(\frac{n}{\epsilon}\ln(\frac{|OPT|}{\epsilon}))$ |
| `stream-c` | $(1-\epsilon)\tau$ | $(1+\alpha)(2/\epsilon+1)|OPT|$ | $O(\log(|OPT|)(\frac{n}{\epsilon}+\mathcal{T}((1+\alpha)|OPT|/\epsilon^2)))$ |

---

**Algorithm 4** `convert`

---

**Input**: An SC instance with threshold $\tau$, a $(\gamma,\beta)$-bicriteria approximation algorithm for SMP, $\alpha > 0$
**Output**: $S \subseteq U$
1: $\kappa \leftarrow (1+\alpha)$, $S \leftarrow \emptyset$.
2: **while** $f(S) < \gamma\tau$ **do**
3:     $S \leftarrow (\gamma,\beta)$-bicriteria approximation for SMP with budget $\kappa$
4:     $\kappa \leftarrow (1+\alpha)\kappa$
5: **return** $S$

---

## 4 Appendix for Section 2

In this portion of the appendix, we present missing details and proofs from Section 2 in the main paper. We first present missing content from Section 2.1 in Section 4.1, followed by missing content from Section 2.2 in Section 4.2, and finally missing content from Section 2.3 is presented in Section 4.3. In this section, we include a more detailed comparison of the algorithms presented in this paper. In addition, pseudocode for the algorithm `convert` of Iyer and Bilmes [2013b] is presented in Algorithm 4.

### 4.1 Additional content to Section 2.1

In this section, we present the detailed statement and proofs of the randomized converting theorem `convert-rand`, and the proofs for the theoretical results of the `thresh-greedy-c` and `stoch-greedy-c` algorithms. The pseudocode for `thresh-greedy-c` is in Algorithm 5. We now provide a proof of Theorem 1.

**Theorem 1.** `thresh-greedy-c` produces a solution with $(\ln(2/\epsilon)+1, 1-\epsilon)$-bicriteria approximation guarantee to MSCP, in $O(\frac{n}{\epsilon}\log(\frac{n}{\epsilon}))$ number of queries of $f$.

*Proof.* Let $r_0$ be defined as $r_0 = \ln(\frac{2}{\epsilon})|OPT|$. By Claim 3.1 in Badanidiyuru and Vondrák [2014], we have that any $s \in U$ that is added to $S$ on Line 6 of Algorithm 5 would satisfy

$$\Delta f(S,s) \geq \frac{1-\epsilon/2}{|OPT|}\sum_{u\in OPT/S}\Delta f(S,u)$$

when it was added. It follows by submodularity that

$$\Delta f(S,s) \geq \frac{1-\epsilon/2}{|OPT|}\sum_{u\in OPT/S}\Delta f(S,u) \geq \frac{1-\epsilon/2}{|OPT|}(f(OPT)-f(S)).$$

And by re-arranging the above equation, and using induction over the elements added to $S$, we have that

$$f(S) \geq \left(1-\left(1-\frac{1-\epsilon/2}{|OPT|}\right)^i\right)f(OPT).$$

Then at the $r_0$-th time of adding new element, it is the case that

$$f(S) \geq \left(1-\left(1-\frac{1-\epsilon/2}{|OPT|}\right)^{r_0}\right)f(OPT) \geq (1-\epsilon)f(OPT) \geq (1-\epsilon)\tau.$$

Thus the condition on Line 3 is satisfied by this point and the algorithm stops after adding the $r_0$-th element. The output solution set satisfies that $|S| \leq |r_0|$.

**Algorithm 5** `thresh-greedy-c`

**Input**: $\epsilon$
**Output**: $S \subseteq U$
1: $S \leftarrow \emptyset$
2: $w = \max_{u \in U} f(u)$
3: **while** $f(S) < (1 - \epsilon)\tau$ **do**
4:      **for** each $u$ in $U$ **do**
5:          **if** $\Delta f(S, u) \geq w$ **then**
6:              $S \leftarrow S \cup \{u\}$
7:      $w = w(1 - \frac{\epsilon}{2})$
8: **return** $S$

---

**Algorithm 6** `convert-rand`

**Input**: A MSCP instance with threshold $\tau$, a $(\gamma, \beta)$-bicriteria approximation algorithm for MSMP where $\gamma$ is in expectation, $\alpha > 0$, $\epsilon > 0$
**Output**: $S \subseteq U$
1: $S_i \leftarrow \emptyset, \forall i \in \{1, ..., \ln(1/\delta)/\ln(\frac{1-\gamma+\epsilon}{1-\gamma})\}$
2: $g \leftarrow (1 + \alpha)$
3: **while** $f(S_i) < (\gamma - \epsilon)\tau \ \forall i$ **do**
4:      **for** $i \in \{1, ..., \ln(1/\delta)/\ln(\frac{1-\gamma+\epsilon}{1-\gamma})\}$ **do**
5:          $S_i \leftarrow (\gamma, \beta)$-bicriteria approximation for MSMP with objective function $f_\tau$ and budget $g$
6:      $g \leftarrow (1 + \alpha)g$
7: **return** $S$

---

We now analyze the number of queries made to $f$. We claim that the algorithm stops before $w = \epsilon \max_{u \in U} f(u)/|OPT|$. This is because at the end of the iteration of the **while** loop in Algorithm 5 corresponding to $w = \epsilon \max_{u \in U} f(u)/|OPT|$, we have that $\Delta f(S, s) < w$ for any $s \in U \setminus S$. Therefore

$$\sum_{s \in OPT/S} \Delta f(S, s) < |OPT/S|w \leq \epsilon \max_{u \in U} f(u) \leq \epsilon f(OPT).$$

It follows that at the end of this iteration of the **while** loop that $f(S) > (1 - \epsilon)f(OPT)$, which means the condition on Line 3 is not satisfied and the algorithm terminates. The number of iterations is thus $O\left(\frac{\ln(|OPT|/\epsilon)}{\epsilon}\right)$ and the query complexity is $O\left(\frac{n}{\epsilon}\ln\left(\frac{|OPT|}{\epsilon}\right)\right)$. $\qquad\square$

Next, we consider the algorithm `convert-rand` that converts algorithms for MSMP to ones for MSCP. Pseudocode for `convert-rand` is provided in Algorithm 6. We now present the proof of Theorem 2.

**Theorem 2.** *Any randomized $(\gamma, \beta)$-bicriteria approximation algorithm for MSMP that runs in time $\mathcal{T}(n)$ where $\gamma$ holds only in expectation can be converted into a $((1 + \alpha)\beta, \gamma - \epsilon)$-bicriteria approximation algorithm for MSCP that runs in time $O(\log_{1+\alpha}(|OPT|)\ln(1/\delta)\mathcal{T}(n))/\ln(\frac{1-\gamma+\epsilon}{1-\gamma}))$ where $\gamma$ holds with probability at least $1 - \delta$.*

*Proof.* Consider the run of the algorithm for MSMP on Line 3 of Algorithm 6 when the parameter $g$ falls into the region $|OPT| \leq g \leq (1 + \alpha)|OPT|$. Notice that it is the case that $f_\tau$ is also monotone and submodular. By the theoretical guarantees of the algorithm for MSMP, we have that for all $i \in \{1, ..., \ln(1/\delta)/\ln(\frac{1-\gamma+\epsilon}{1-\gamma})\}$

$$\mathbb{E}f_\tau(S_i) \geq \gamma \max_{|X| \leq g}\{f_\tau(X)\} \geq \gamma \max_{|X| \leq |OPT|}\{f_\tau(X)\} \geq \gamma\tau.$$

From Markov's inequality, we have that for each $i \in \{1, ..., \ln(1/\delta)/\ln(\frac{1-\gamma+\epsilon}{1-\gamma})\}$

$$P(f_\tau(S_i) \leq (\gamma - \epsilon)\tau) \leq P(\tau - f_\tau(S_i) > \frac{1 - \gamma + \epsilon}{1 - \gamma}(\tau - \mathbb{E}f_\tau(S_i))) \leq \frac{1 - \gamma}{1 + \epsilon - \gamma}.$$

---
**Algorithm 7** `stoch-bi`

---
**Input:** $\epsilon > 0$
**Output:** $S \subseteq U$
1: $S \leftarrow \emptyset$
2: **while** $|S| < \ln(\frac{3}{2\epsilon})\kappa$ **do**
3: $\quad R \leftarrow$ sample $\frac{n}{\kappa}\ln(\frac{3}{2\epsilon})$ elements from $U$
4: $\quad u \leftarrow \text{argmax}_{x \in R}\Delta f(S, x)$
5: **return** $S$

---

Then the probability that none of the subsets $S_i$ can reach the stopping condition can be bounded by

$$P(f(S_i) \le (\gamma - \epsilon)\tau, \forall i) = P(f_\tau(S_i) \le (\gamma - \epsilon)\tau, \forall i)$$

$$= \prod_{i=1}^{\ln(1/\delta)/\ln(\frac{1-\gamma+\epsilon}{1-\gamma})} P(f_\tau(S_i) \le (\gamma - \epsilon)\tau)$$

$$\le (\frac{1-\gamma}{1+\epsilon-\gamma})^{\ln(1/\delta)/\ln(\frac{1-\gamma+\epsilon}{1-\gamma})} = \delta.$$

This means with probability at least $1 - \delta$, `convert-rand` stops when $g$ reaches the region where $|OPT| \le g \le (1 + \alpha)|OPT|$ since the condition of the **while** loop is not satisfied. Therefore, by the assumption that the subroutine algorithm is a $(\gamma, \beta)$-bicriteria approximation algorithm, we have that the output solution $S$ satisfies that $|S| \le \beta g \le \beta(1 + \alpha)|OPT|$. It also implies that there are at most $O(\log_{1+\alpha}|OPT|)$ number of guesses of the cardinality of the optimal solution. Since for each guess, we run the MSMP for $\ln(1/\delta)/\ln(\frac{1-\gamma+\epsilon}{1-\gamma})$ times, the algorithm runs in time $O(\log_{1+\alpha}(|OPT|)\ln(1/\delta)\mathcal{T}(n)/\ln(\frac{1-\gamma+\epsilon}{1-\gamma}))$.

$\square$

Before we present the proof of the theoretical guarantees of the algorithm `stoch-greedy-c`, we first introduce an algorithm for MSMP with bicriteria approximation guarantee called `stoch-bi`. `stoch-bi` is an extension of the stochastic greedy algorithm of Mirzasoleiman et al. [2015a], which produces an infeasible solution to the instance of MSMP that has $f$ value arbitrarily close to that of the optimal solution. Pseudocode for `stoch-bi` is provided in Algorithm 7. Notice that if we use the `stoch-bi` as a subroutine for `convert-rand`, we can obtain an algorithm for MSCP that runs in $O(n \ln^2(3/\epsilon)\ln(1/\delta)\log_{1+\alpha}|OPT|)$, which is less queries than the standard greedy algorithm, but worse than our algorithm `stoch-greedy-c` by a factor of $(1 + \alpha)/\alpha$.

**Theorem 7.** *`stoch-bi` produces a solution with $(1-\epsilon, \ln(\frac{3}{2\epsilon}))$-bicriteria approximation guarantee to MSMP, where the $1 - \epsilon$ holds in expectation, in $O(n \ln^2(\frac{3}{2\epsilon}))$ queries of $f$.*

*Proof.* Our argument to prove Theorem 7 follows a similar approach as Mirzasoleiman et al. [2015a]. Let the partial solution $S$ before $i$-th iteration of the **while** loop of Algorithm 7 be $S_i$, the item that is added to set $S$ during iteration $i$ be $u_i$, and the sampled set $R$ during iteration $i$ be $R_i$. Consider the beginning of any iteration $i$ of the **while** loop. Then because of the greedy choice for $u_i$ it is the case that if $R_i \cap OPT \ne \emptyset$,

$$\Delta f(S_i, u_i) \ge \Delta f(S_i, x)$$

for all $x \in R_i \cap OPT$. Therefore we have that in expectation over the randomly chosen set $R_i$

$$\mathbb{E}[\Delta f(S_i, u_i)|S_i] \ge \mathbf{Pr}(R_i \cap OPT \ne \emptyset)\mathbb{E}[\Delta f(S_i, u_i)|S_i, R_i \cap OPT \ne \emptyset]$$

$$\ge \mathbf{Pr}(R_i \cap OPT \ne \emptyset)\mathbb{E}[\frac{1}{|R_i \cap OPT|}\sum_{x \in R_i \cap OPT}\Delta f(S_i, x)|S_i, R_i \cap OPT \ne \emptyset]$$

$$= \mathbf{Pr}(R_i \cap OPT \ne \emptyset)\frac{1}{\kappa}\sum_{x \in OPT}\Delta f(S_i, x) \tag{1}$$

where the last inequality follows since the elements of $R_i$ are chosen uniformly randomly, implying that all elements of $OPT$ are equally likely to be chosen.

Let the number of samples $s = \frac{n}{k} \ln(\frac{3}{2\epsilon})$. The probability $\mathbf{Pr}(R_i \cap OPT \neq \emptyset)$ can be lower bounded as

$$
\begin{aligned}
\mathbf{Pr}(R_i \cap OPT \neq \emptyset) &= 1 - \mathbf{Pr}(R_i \cap OPT = \emptyset) \\
&= 1 - \frac{\binom{n-\kappa}{s}}{\binom{n}{s}} \\
&\geq 1 - (\frac{n-\kappa}{n})^s \\
&\geq 1 - \frac{2\epsilon}{3},
\end{aligned}
\tag{2}
$$

where the last inequality comes from the fact that $(1 - \frac{1}{x})^x \leq e^{-1}$. By submodularity, we have

$$
\sum_{u \in OPT} \Delta f(S_i, u) \geq f(OPT) - f(S_i).
\tag{3}
$$

Plug Equations (2) and (3) into (1) yields that

$$
\mathbb{E}[\Delta f(S_i, u_i)|S_i] \geq \frac{1 - 2\epsilon/3}{\kappa}(f(OPT) - f(S_i)).
\tag{4}
$$

Therefore,

$$
\mathbb{E}[f(S_{i+1})|S_i] \geq \frac{1 - 2\epsilon/3}{\kappa} f(OPT) + (1 - \frac{1 - 2\epsilon/3}{\kappa}) f(S_i)).
$$

By induction and by taking expectation over $S_i$, we can get that at the last iteration

$$
\begin{aligned}
\mathbb{E}[f(S)] &\geq \frac{1 - 2\epsilon/3}{\kappa} \sum_{i=0}^{|S|-1} (1 - \frac{1 - 2\epsilon/3}{\kappa})^i f(OPT) \\
&\geq \{1 - (1 - \frac{1 - 2\epsilon/3}{\kappa})^{|S|}\} f(OPT) \\
&\geq (1 - (2\epsilon/3)^{1-2\epsilon/3}) f(OPT) \geq (1 - \epsilon) f(OPT),
\end{aligned}
$$

where in the last inequality, we use the fact that $|S| = \ln(3/2\epsilon)\kappa$. $\qquad\square$

We now present the omitted proofs for Lemmas 1, 2, and 3 about our algorithm `stoch-greedy-c`. Once these lemmas are proven, we next prove Theorem 3 using the lemmas.

**Lemma 1.** *Define $\ell$ to be the integer such that $(1+\alpha)^{\ell-1} \leq |OPT| \leq (1+\alpha)^\ell$. Consider any of the sets $S_i$ at the beginning of an iteration on Line 4 where $g \leq (1+\alpha)^\ell$. Then if $u_i$ is the random element that will be added on Line 6, we have that*

$$
\mathbb{E}[\Delta f_\tau(S_i, u_i)|S_i] \geq \frac{1 - \epsilon/3}{(1+\alpha)^\ell}(\tau - f_\tau(S_i)) \geq \frac{1 - \epsilon/3}{(1+\alpha)|OPT|}(\tau - f_\tau(S_i)).
$$

*Proof.* Define $OPT^* = \arg\max_{|X| \leq (1+\alpha)^\ell} f(X)$. Then we have $f(OPT^*) \geq f(OPT) \geq \tau$. Therefore $f_\tau(OPT^*) \geq \tau$. Consider an iteration of the **while** loop on Line 4 of `stoch-greedy-c` where $g \leq (1+\alpha)^\ell$, and some iteration of inside **for** loop corresponding to set $S_i$. From Line 5 of `stoch-greedy-c`, we have that the size of the randomly sampled subset $R_i$ for $S_i$ is $\min\{n, n \ln(3/\epsilon)/g\}$. Then if $n \ln(3/\epsilon)/g < n$, we have that

$$
|R_i| = \frac{n}{g} \ln(3/\epsilon) \geq \frac{n}{(1+\alpha)^\ell} \ln(3/\epsilon).
$$

This implies that if $n \ln(3/\epsilon)/g < n$, when we randomly sample $R_i$

$$
P(R_i \cap OPT^* = \emptyset) = (1 - |OPT^*|/n)^{|R_i|} \leq (1 - (1+\alpha)^\ell/n)^{\frac{n \ln(3/\epsilon)}{(1+\alpha)^\ell}} \leq \epsilon/3.
$$

On the other hand, if $n \ln(3/\epsilon)/g \geq n$, then $P(R_i \cap OPT^* = \emptyset) = 0$ and the above inequality also holds.

Notice that $f_\tau$ is monotone and submodular. Let $u_i$ be the elements of $R_i$ which will be added to $S_i$. Then

$$\mathbb{E}[\Delta f_\tau(S_i, u_i)|R_i \cap OPT^* \neq \emptyset] \geq \frac{1}{|OPT^*|} \sum_{o \in OPT^*} \Delta f_\tau(S_i, o)$$

$$\overset{(i)}{\geq} \frac{1}{|OPT^*|}(f_\tau(OPT^*) - f_\tau(S_i))$$

$$\geq \frac{1}{(1+\alpha)^\ell}(\tau - f_\tau(S_i)).$$

where (i) is implied by the fact that $f_\tau$ is monotone and submodular. Altogether we have that

$$\mathbb{E}[\Delta f_\tau(S_i, u_i)|S_i] \geq P(R \cap OPT^* \neq \emptyset)\mathbb{E}[\Delta f_\tau(S_i, u_i)|R \cap OPT^* \neq \emptyset]$$

$$\geq \frac{1 - \epsilon/3}{(1+\alpha)^\ell}(\tau - f_\tau(S_i)).$$

□

**Lemma 2.** *Once $r \geq (1+\alpha)\ln(3/\epsilon)|OPT|$, we have that $\mathbb{E}[f_\tau(S_i)] \geq \left(1 - \frac{\epsilon}{2}\right)\tau$ for all $i$.*

*Proof.* Define $\ell$ to be the integer such that $(1+\alpha)^{\ell-1} \leq |OPT| \leq (1+\alpha)^\ell$. Then notice that throughout stoch-greedy-c until $r$ is incremented to be $\ln(3/\epsilon)(1+\alpha)^\ell$, it is the case that $g \leq (1+\alpha)^\ell$. Therefore Lemma 1 applies for all marginal gains of adding elements to the sets until the end of the $\ln(3/\epsilon)(1+\alpha)^\ell$ iteration of the **while** loop. Consider any of the sets $S_i$. By applying recursion to Lemma 1, we have that at the end of any iteration $r \leq \ln(3/\epsilon)(1+\alpha)^\ell$ of the **while** loop

$$\mathbb{E}[f_\tau(S_i)] \geq \left(1 - \left(1 - \frac{1 - \epsilon/3}{(1+\alpha)^\ell}\right)^r\right)\tau.$$

Therefore, Once $r$ reaches $\ln(3/\epsilon)(1+\alpha)^\ell$, the expectation of $S_i$ is

$$\mathbb{E}[f_\tau(S_i)] \geq \left(1 - \left(1 - \frac{1 - \epsilon/3}{(1+\alpha)^\ell}\right)^{(1+\alpha)^\ell \ln(3/\epsilon)}\right)\tau \geq \left(1 - \frac{\epsilon}{2}\right)\tau.$$

Because $\ln(3/\epsilon)(1+\alpha)^\ell \leq (1+\alpha)\ln(3/\epsilon)|OPT|$ and $f$ is monotonic, Lemma 2 holds. □

**Lemma 3.** *With probability at least $1 - \delta$, once $r \geq (1+\alpha)\ln(3/\epsilon)|OPT|$, we have that $\max_i f(S_i) \geq (1-\epsilon)\tau$.*

*Proof.* Since $f_\tau(S) = \min\{f(S), \tau\}$, we have that $\tau - f_\tau(S) \in [0, \tau]$ for any $S \in U$. By applying the Markov's inequality, we have that

$$P(\tau - f_\tau(S_j) \geq 2\mathbb{E}(\tau - f_\tau(S_j)) \leq \frac{1}{2}$$

From Lemma 2, we have $\mathbb{E}(\tau - f_\tau(S_j)) \leq \frac{\epsilon\tau}{2}$. Then

$$P(f_\tau(S_j) \leq (1-\epsilon)\tau) = P(\tau - f_\tau(S_j) \geq \epsilon\tau)$$

$$\leq P(\tau - f_\tau(S_j) \geq 2\mathbb{E}(\tau - f_\tau(S_j)))$$

$$\leq \frac{1}{2}.$$

Thus we have

$$P(\max_j f_\tau(S_j) > (1-\epsilon)\tau) = 1 - P(f_\tau(S_j) \leq (1-\epsilon)\tau, \forall j)$$

$$= 1 - P(f_\tau(S_1) \leq (1-\epsilon)\tau)^{\ln(1/\delta)/\ln(2)}$$

$$\geq 1 - \left(\frac{1}{2}\right)^{\ln(1/\delta)/\ln(2)} \geq 1 - \delta.$$

□

**Theorem 3.** *Suppose that* `stoch-greedy-c` *is run for an instance of MSCP. Then with probability at least* $1 - \delta$, `stoch-greedy-c` *outputs a solution $S$ that satisfies a $((1 + \alpha) \ln(3/\epsilon), 1 - \epsilon)$-bicriteria approximation guarantee in at most*

$$O\left( \frac{\alpha}{1 + \alpha} n \ln(1/\delta) \ln^2(3/\epsilon) \log_{1+\alpha}(|OPT|) \right)$$

*queries of $f$.*

*Proof.* From Lemma 3, we have that the algorithm stops by the time $r$ reaches $(1+\alpha) \ln(3/\epsilon)|OPT|$ with probability at least $1 - \delta$. Before $r$ reaches this point, consider the number of queries made to $f$ over the duration that $g$ is a certain value $(1 + \alpha)^m$. Then for each of the $O(\ln(1/\delta))$ sets a total of at most

$$\left( \ln\left(\frac{3}{\epsilon}\right) (1 + \alpha)^m - \ln\left(\frac{3}{\epsilon}\right)(1 + \alpha)^{m-1} \right) \frac{n \ln(\frac{3}{\epsilon})}{(1 + \alpha)^m} = \frac{\alpha}{1 + \alpha} n \ln^2(3/\epsilon)$$

queries are made to $f$. Because $g$ takes on at most $O(\log_{1+\alpha}(|OPT|))$ values before $r$ reaches $(1 + \alpha) \ln(3/\epsilon)|OPT|$, Theorem 3 follows. $\square$

## 4.2 Additional content for Section 2.2

In this section, we present the proofs of theoretical results from Section 2.2. First, in Claim 1 we present a useful claim from Crawford [2023] about submodular functions in order to prove Lemma 4. Next, we use Claim 1 in order to prove Lemma 4. Finally, Lemma 4 is used to prove the main result for `stream-c`, Theorem 4.

**Claim 1** (Claim 1 in Crawford [2023]). *Let $A_1, ..., A_m \subseteq U$ be disjoint, and $B \subseteq U$. Then there exists $i \in \{1, ..., m\}$ such that $f(A_i \cup B) \geq (1 - 1/m)f(B)$.*

**Lemma 4.** *By the time that $g$ reaches the region $[|OPT|, (1 + \alpha)|OPT|]$ and the loop on Line 4 of* `stream-c` *has completed, there exists a set $X \subseteq \cup S_i$ of size at most $(2/\epsilon + 1)g$ such that $f(X) \geq (1 - \epsilon)\tau$.*

*Proof.* Suppose that `stream-c` has reached the end of the loop on Line 4 when $g \geq |OPT|$. We first consider the case where there exists some set $S_t$ with $t \in \{1, ..., 2/\epsilon\}$ such that $|S_t| = 2g/\epsilon$. In this case it follows that $f(S_t) = \sum_{i=1}^{2g/\epsilon} \Delta f(S_t^i, u_i) \geq \tau$, where $S_t^i$ is the set of $S_t$ before adding the $i$-th element $u_i$, and so the Lemma statement is proven.

Next, we consider the case where $|S_j| < 2g/\epsilon, \forall j \in [2/\epsilon]$. Let $OPT_1 = OPT \cap (\cup_{i=1}^{2/\epsilon} S_i)$ and $OPT_2 = OPT/OPT_1$. By Claim 1, there exists a set $S_t$ such that

$$f(S_t \cup OPT) \geq (1 - \epsilon/2)f(OPT) \geq (1 - \epsilon/2)\tau.$$

Since $|S_t| < 2g/\epsilon$ at the end of the algorithm, we can see each element $o$ in $OPT_2$ is not added into $S_t$ because $\Delta f(S_t, o) \leq \epsilon\tau/(2g)$ at the time $o$ is seen in the loop on Line 4. By submodularity and the fact that $g \geq |OPT| \geq |OPT_2|$,

$$\sum_{o \in OPT_2} \Delta f(S_t \cup OPT_1, o) < \epsilon|OPT_2|\tau/(2g) \leq \epsilon\tau/2.$$

Therefore

$$\begin{aligned}
(1 - \epsilon/2)\tau &\leq f(S_t \cup OPT) \\
&\leq f(S_t \cup OPT_1) + \Delta f(S_t \cup OPT_1, OPT_2) \\
&\leq f(S_t \cup OPT_1) + \sum_{o \in OPT_2} \Delta f(S_t \cup OPT_1, o) \\
&\leq f(S_t \cup OPT_1) + \epsilon\tau/2.
\end{aligned}$$

Therefore $f(S_t \cup OPT_1) \geq (1 - \epsilon)\tau$. Because $S_t \cup OPT_1 \subseteq \cup S_i$, and it is the case that $|S_t \cup OPT_1| \leq |S_t| + |OPT_1| \leq 2g/\epsilon + g$, so the Lemma statement is proven. $\square$

**Theorem 4.** *Suppose that* `stream-c` *is run for an instance of SCP. Then* `stream-c` *returns* $S$ *such that* $f(S) \geq (1 - \epsilon)\tau$ *and* $|S| \leq (1 + \alpha)(2/\epsilon + 1)|OPT|$ *in at most*

$$\log_{1+\alpha}(|OPT|)\left(\frac{2n}{\epsilon} + \mathcal{T}\left((1 + \alpha)\left(\frac{4}{\epsilon^2}|OPT|\right)\right)\right)$$

*queries of* $f$, *where* $\mathcal{T}(m)$ *is the number of queries to* $f$ *of the algorithm for SMP used on Line 7 of Algorithm 2 on an input set of size* $m$.

*Proof.* By Lemma 4, once `stream-c` reaches $g \geq |OPT|$ then $f(S) \geq (1 - \epsilon)\tau$ will be satisfied and therefore the **while** loop will exit. This implies that $g \leq (1 + \alpha)|OPT|$ once `stream-c` exits, and therefore $|S| \leq (2/\epsilon + 1)g \leq (1 + \alpha)(2/\epsilon + 1)|OPT|$. Therefore the qualities of $S$ stated in Theorem 4 are proven. As far as the number of queries of $f$, it takes at most $\log_{1+\alpha}(|OPT|)$ iterations of the loop to reach the point that $g \geq |OPT|$. In addition, each iteration of the loop makes $\frac{2n}{\epsilon} + \mathcal{T}\left((1 + \alpha)\left(\frac{4}{\epsilon^2}|OPT|\right)\right)$ queries of $f$. Therefore the bound on the number of queries of $f$ in Theorem 4 is proven. □

### 4.3 Supplementary material to Section 2.3

We now present additional theoretical details for Section 2.3, where we consider RMSCP. First, we prove Theorem 5 about our algorithm `convert-reg` which converts algorithms for RMSMP into ones for RMSCP.

**Theorem 5.** *Suppose that we have an algorithm* `reg` *for maximization of a regularized submodular function subject to a cardinality constraint* $\kappa$, *and that algorithm is guaranteed to return a set* $S$ *of cardinality at most* $\rho\kappa$ *such that*

$$g(S) - c(S) \geq \gamma g(X) - \beta c(X)$$

*for all* $X$ *such that* $|X| \leq \kappa$ *in time* $T(n)$. *Then the algorithm* `convert-reg` *using* `reg` *as a subroutine returns a set* $S$ *such that*

$$g(S) - \frac{\gamma}{\beta}c(S) \geq \gamma\tau$$

*and* $|S| \leq (1 + \alpha)\rho|OPT|$ *in time* $O(\log_{1+\alpha}(n)T(n))$.

*Proof.* Let $OPT$ be the optimal solution to the instance of RMSCP. Consider the iteration of `convert-reg` where $\kappa$ has just increased above $|OPT|$, i..e $|OPT| \leq \kappa \leq (1 + \alpha)|OPT|$. Then we run `reg` with input objective $g - \frac{\gamma}{\beta}c$ and budget $\kappa \geq |OPT|$. Then by the assumptions on `reg` we have that

$$g(S) - \frac{\gamma}{\beta}c(S) \geq \gamma(g(OPT) - c(OPT)) \geq \gamma\tau$$

and $|S| \leq \rho\kappa \leq (1 + \alpha)\rho|OPT|$. □

We now fill in the missing information concerning our algorithm `distorted-bi`. Recall the definition of

$$\Phi_i(X) = \left(1 - \frac{1}{\kappa}\right)^{t-i} g(X) - c(X),$$

where $t = \ln(1/\epsilon)\kappa$, which is used in both the pseudocode for `distorted-bi` and throughout the proofs. First, pseudocode for `distorted-bi` is presented in Algorithm 8. Next, we present and prove Lemma 5, and then use Lemma 5 in order to prove our main result for `distorted-bi`, Theorem 6.

**Lemma 5.** *Consider any iteration* $i + 1$ *of the **while** loop in* `distorted-bi`. *Let* $S_i$ *be defined to be* $S$ *after the* $i$-th *iteration of the **while** loop in* `distorted-bi`. *Then for any* $X \subseteq U$ *such that* $|X| \leq \kappa$,

$$\Phi_{i+1}(S_{i+1}) - \Phi_i(S_i) \geq \frac{1}{\kappa}\left(1 - \frac{1}{\kappa}\right)^{t-(i+1)} g(X) - \frac{c(X)}{\kappa}.$$

**Algorithm 8** `distorted-bi`

---

**Input**: $\epsilon$
**Output**: $S \subseteq U$
 1: $S \leftarrow \emptyset$
 2: $i \leftarrow 1$
 3: **while** $|S| < \ln(1/\epsilon)\kappa$ **do**
 4:     $u \leftarrow \text{argmax}_{x \in U} \Delta\Phi_i(S, x)$
 5:     **if** $\Delta\Phi_i(S, x) > 0$ **then**
 6:         $S \leftarrow S \cup \{u\}$
 7:         $i \leftarrow i + 1$
 8:     **else**
 9:         **break**
10: **return** $S$

---

*Proof.* First, suppose that during iteration $i + 1$ an element $s_{i+1}$ was added to $S$. First, notice that

$$\Phi_{i+1}(S_{i+1}) - \Phi_i(S_i) = \Phi_{i+1}(S_{i+1}) - \Phi_{i+1}(S_i) + \Phi_{i+1}(S_i) - \Phi_i(S_i)$$

$$= \Delta\Phi_{i+1}(S_i, s_{i+1}) + \frac{1}{\kappa}\left(1 - \frac{1}{\kappa}\right)^{t-(i+1)} g(S_i). \tag{5}$$

In addition, notice that for any subset of $U$ such that $|X| \leq \kappa$

$$\Delta\Phi_{i+1}(S_i, s_{i+1}) \overset{(a)}{\geq} \frac{1}{\kappa} \sum_{o \in X} (\Delta\Phi_{i+1}(S_i, o))$$

$$\geq \frac{1}{\kappa} \sum_{o \in X} \left(\left(1 - \frac{1}{\kappa}\right)^{t-(i+1)} \Delta g(S_i, o) - \Delta c(S_i, o)\right)$$

$$\geq \frac{1}{\kappa}\left(1 - \frac{1}{\kappa}\right)^{t-(i+1)} \sum_{o \in X} \Delta g(S_i, o) - \frac{c(X)}{\kappa}$$

$$\overset{(b)}{\geq} \frac{1}{\kappa}\left(1 - \frac{1}{\kappa}\right)^{t-(i+1)} (g(S_i \cup X) - g(S_i)) - \frac{c(X)}{\kappa}$$

$$\overset{(c)}{\geq} \frac{1}{\kappa}\left(1 - \frac{1}{\kappa}\right)^{t-(i+1)} (g(X) - g(S_i)) - \frac{c(X)}{\kappa}$$

where (a) is because of the greedy choice of $s_i$; (b) is by the submodularity of $g$; and (c) is by the monotonicity of $g$. Combining the previous two equations gives the result of Lemma 5.

Next, we consider the case where $\Delta\Phi_i(S_i, x) \leq 0$ for all $x \in U$, and so there is no new element added. In this case, we can randomly choose an element from the current solution $S_i$ as the added element $s_{i+1}$. It is not hard to verify that the above two inequalities still hold in this case, and therefore again we have the result of Lemma 5. $\qquad\square$

**Theorem 6.** *Suppose that* `distorted-bi` *is run for an instance of RMSMP. Then* `distorted-bi` *produces a solution $S$ in $O(n\kappa \ln(1/\epsilon))$ queries of $f$ such that $|S| \leq \ln(1/\epsilon)\kappa$ and for all $X \subseteq U$ such that $|X| \leq \kappa$, $g(S) - c(S) \geq (1 - \epsilon)g(X) - \ln(1/\epsilon)c(X)$.*

*Proof.* We will use Lemma 5 to prove Theorem 6. Define $t = \ln(1/\epsilon)\kappa$. Then we see that

$$
g(S_t) - c(S_t) \overset{(a)}{\geq} \Phi_t(S_t) - \Phi_0(S_0)
$$

$$
= \sum_{i=0}^{t-1} (\Phi_{i+1}(S_{i+1}) - \Phi_i(S_i))
$$

$$
\overset{(b)}{\geq} \sum_{i=0}^{t-1} \left( \frac{1}{\kappa} \left( 1 - \frac{1}{\kappa} \right)^{t-(i+1)} g(X) - \frac{c(X)}{\kappa} \right)
$$

$$
= \frac{g(X)}{\kappa} \sum_{i=0}^{t-1} \left( 1 - \frac{1}{\kappa} \right)^{t-(i+1)} - \ln(1/\epsilon)\, c(X)
$$

$$
= \frac{g(X)}{\kappa} \sum_{i=0}^{t-1} \left( 1 - \frac{1}{\kappa} \right)^{i} - \ln(1/\epsilon)\, c(X)
$$

$$
\overset{(c)}{=} \left( 1 - \left( 1 - \frac{1}{\kappa} \right)^{t} \right) g(X) - \ln(1/\epsilon)\, c(X)
$$

$$
\geq (1 - \epsilon)g(X) - \ln(1/\epsilon)\, c(X)
$$

where (a) is because $g(\emptyset) \geq 0$; (b) is because Lemma 5; and (c) is by the formula for geometric series. $\qquad\square$

## 5 Supplementary material for Section 3

In this section, we present supplementary material to Section 3. In particular, we present additional details about the experimental setup in Section 5.1, and additional experimental results in Section 5.2.

### 5.1 Experimental setup

First of all, we provide more details about the two applications used to evaluate the algorithms proposed in the main paper. For MSCP, the application considered here is data summarization, where $f$ is a function that represents how well a subset could summarize the whole dataset. The problem definition is as follows.

**Definition 1.** *(**Data Summarization**) Suppose there are a total of $n$ elements denoted as $U$. Let $T$ be a set of tags. Each element in $U$ is tagged with a set of elements from $T$ via function $t : U \rightarrow 2^T$. The function $f$ is defined as*

$$
f(S) = |\cup_{s \in S} t(s)|, \qquad \forall S \in U.
$$

From the definition, we can see that $f$ is both monotone and submodular. For general SCP, where $f$ can be nonmonotone, the application we consider is where $f$ is a graph cut function, which is a submodular but not necessarily monotone function.

**Definition 2.** *(**Graph cut**) Let $G = (V, E)$ be a graph, and $w : E \rightarrow \mathbb{R}_{\geq 0}$ be a function that assigns a weight for every edge in the graph. The function $f : 2^V \rightarrow \mathbb{R}_{\geq 0}$ maps a subset of vertices $X \subseteq V$ to the total weight of edges between $X$ and $V \backslash X$. More specifically,*

$$
f(X) = \sum_{x \in X, y \in V \backslash X} w(x, y).
$$

In addition to the datasets considered in the main paper, we also look at graph cut instances on the com-Amazon ($n = 334863$, 925872 edges), ego-Facebook ($n = 4039$, 88234 edges), and email-Enron ($n = 36692$, 183831 edges) graphs from the SNAP large network collection [Leskovec and Sosič, 2016].

We now present more details about the algorithms "EX" and "F-EX". EX begins by using a greedy algorithm to find a solution that meets the desired threshold. If this greedy choice fails, EX begins a

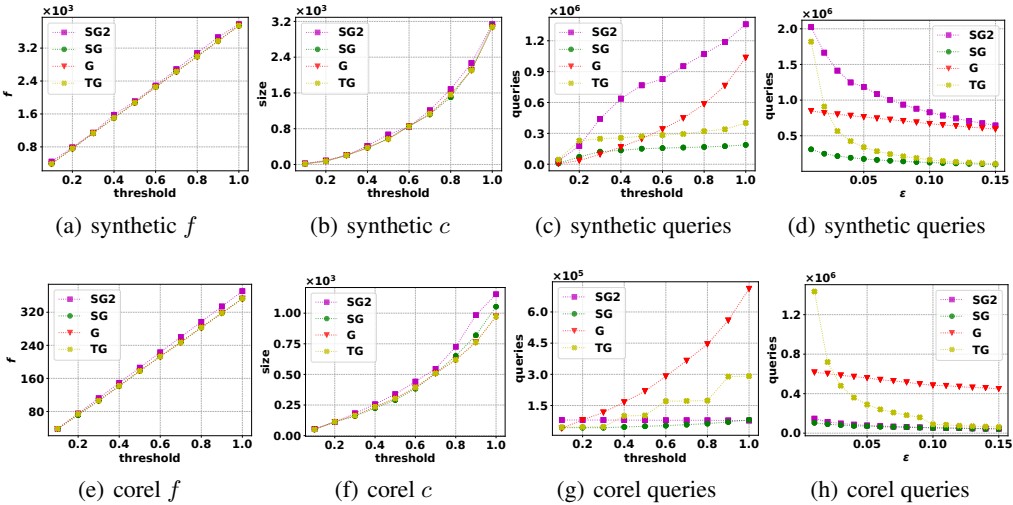

|     |     |     |     |
| --- | --- | --- | --- |
| (a) synthetic $f$ | (b) synthetic $c$ | (c) synthetic queries | (d) synthetic queries |
| (e) corel $f$ | (f) corel $c$ | (g) corel queries | (h) corel queries |

Figure 2: The experimental results of running different greedy algorithms on the instances of data summarization on the synthetic dataset, and corel dataset.

search of all feasible sets where each element considered is in order of decreasing marginal gains. Lazy updates are used for computing marginal gains. "F-EX" uses a similar approach as EX, but as described in the main paper limits the search to a smaller portion of the elements. In particular, as described in the main paper, the F-EX algorithm first checks if the optimization problem in line 7 of `stream-c` is an unconstrained SMP. If so, the algorithm adds all monotone elements from the union of $\{S_1, ...S_{2/\epsilon}\}$ to the solution set and then explores the non-monotone elements in the rest of $\cup_{i=1}^{2/\epsilon} S_i$.

## 5.2 Additional experimental results

First of all, we present some additional monotone experimental results on the synthetic data and the corel dataset. The corel dataset is the Corel5k set of images in Duygulu et al. [2002] ($n = 4500$). The synthetic data we used here is a data summarization instance with a total of $m = 4000$ elements and $2000$ ($n = 2000$) subsets of elements. Each subset is randomly generated in the following way: let us denote the set of elements as $[m] = \{1, 2, 3..., m\}$, each element $i \leq 250$ is added to the subset with probability $0.4$, and each element $250 < i \leq m$ is added with probability $0.002$. The four algorithms examined here can be found in Section 3. We compare the four algorithms for different values of $\tau$ and $\epsilon$. When $\epsilon$ is varied, $\tau$ is fixed at $0.9f(U)$ for corel dataset and synthetic dataset with $U$ being the universe of the two instances respectively. When $\tau$ is varied, $\epsilon$ is fixed at $0.05$. The results on the synthetic data are presented in Figure 2(a), 2(b), 2(c), and 2(d). The results on the corel dataset are plotted in Figure 2(e), 2(f), 2(g) and 2(h). From the results, we can see that the results on the corel dataset are in line with the results in the main paper. However, on the synthetic dataset, it is worth noting that the "SG" (`stoch-greedy-c`) algorithm requires fewer queries compared with the other three algorithms even when $\epsilon$ is large, which further demonstrates the advantages of our algorithms.

The additional experiments we present are further exploration of our algorithm `stream-c`. In Figure 3, we present additional experimental results analogous to those presented in the main paper for graph cut, but on the additional datasets described above. In summary, we see many of the same patterns exhibited in these instances as discussed in the main paper. In addition, we include an additional set of experiments analyzing how many non-monotone elements there are in the union of $\{S_1, ...S_{2/\epsilon}\}$ in `stream-c` in order to analyze how effective we would expect F-EX to be. The number ("num") and portion ("pt") of the nonmonotone elements of the union of $\{S_1, ...S_{2/\epsilon}\}$ on the instances with the graph cut objective are plotted in Figures 4 and 5. If that instance did not require an exact search (meaning that the initial greedy heuristic found a solution), then -1 is plotted. From the figures, we can see that in most cases, the number of nonmonotone elements is either $0$ or very small, which explains why the fast exact algorithm requires fewer queries than the exact algorithm in these cases.

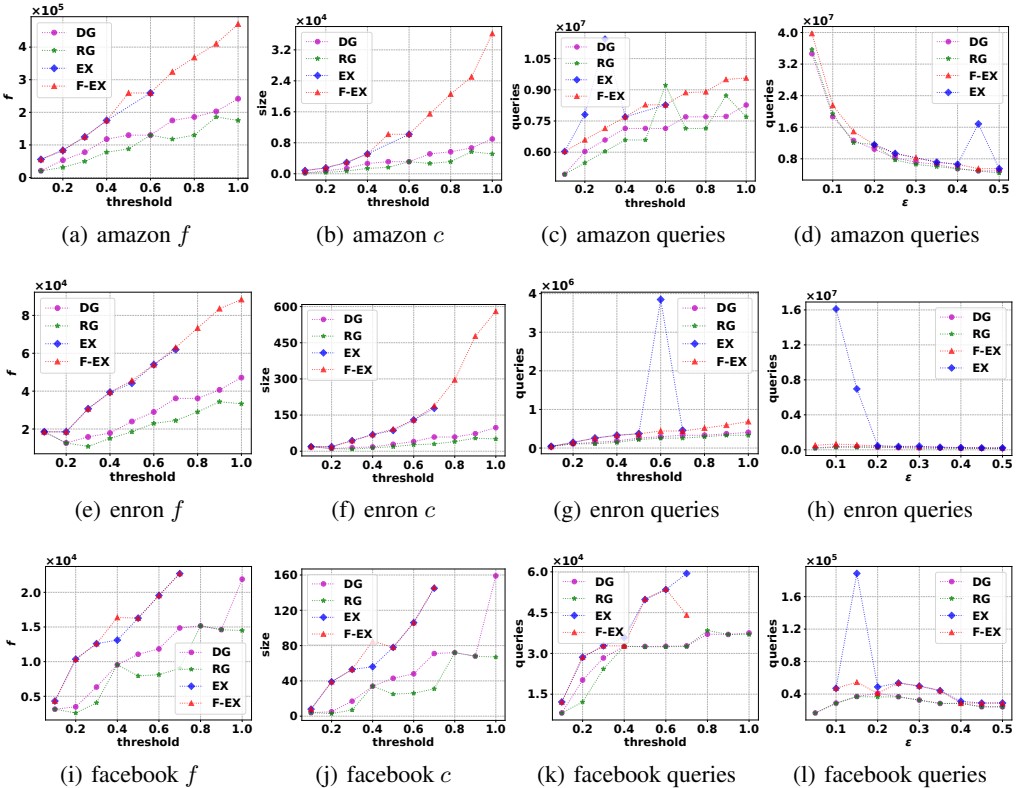

Figure 3: The experimental results of running `stream-c` on the instances of graph cut on the com-Amazon graph ("amazon"), email-Enron ("enron") and ego-Facebook ("facebook") dataset.

This implies that in many instances of SCP, `stream-c` is able to cut down the original instance to one that is nearly monotone and much easier to solve.

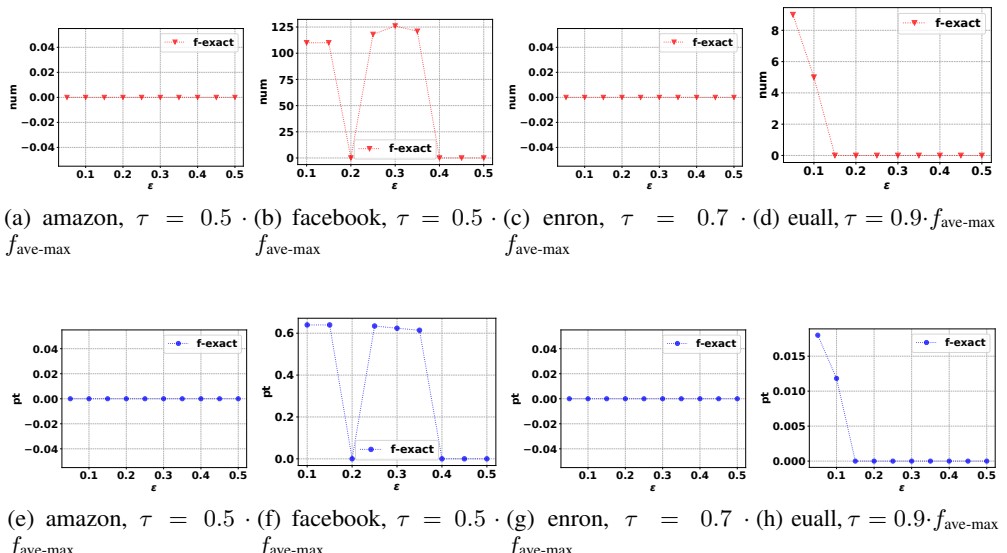

(a) amazon, $\tau = 0.5 \cdot f_{\text{ave-max}}$ (b) facebook, $\tau = 0.5 \cdot f_{\text{ave-max}}$ (c) enron, $\tau = 0.7 \cdot f_{\text{ave-max}}$ (d) euall, $\tau = 0.9 \cdot f_{\text{ave-max}}$

(e) amazon, $\tau = 0.5 \cdot f_{\text{ave-max}}$ (f) facebook, $\tau = 0.5 \cdot f_{\text{ave-max}}$ (g) enron, $\tau = 0.7 \cdot f_{\text{ave-max}}$ (h) euall, $\tau = 0.9 \cdot f_{\text{ave-max}}$

Figure 4: The second experiments. We plot the number and portion of the non-monotone elements. Here $x$-axis refers to values of $\tau$.

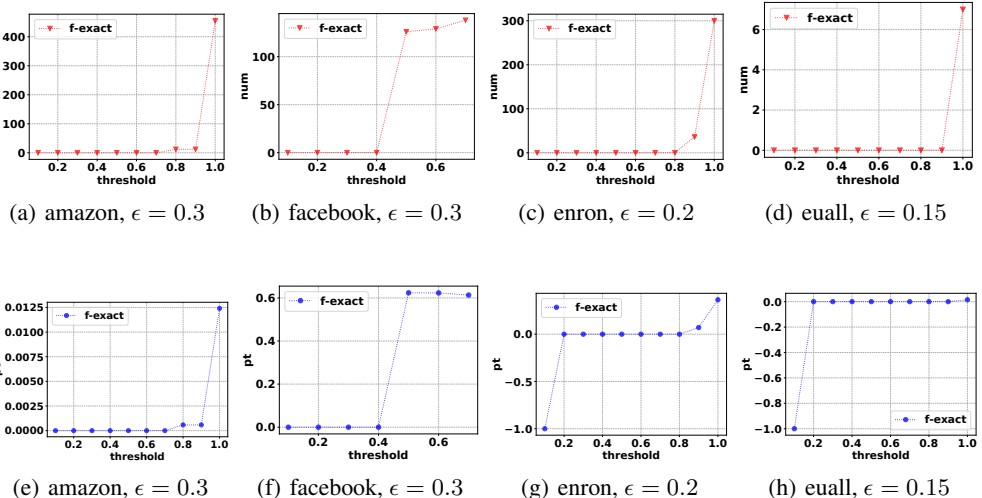

(a) amazon, $\epsilon = 0.3$ (b) facebook, $\epsilon = 0.3$ (c) enron, $\epsilon = 0.2$ (d) euall, $\epsilon = 0.15$

(e) amazon, $\epsilon = 0.3$ (f) facebook, $\epsilon = 0.3$ (g) enron, $\epsilon = 0.2$ (h) euall, $\epsilon = 0.15$

Figure 5: The second experiments. We plot the number and portion of the non-monotone elements. Here $x$-axis refers to values of $\epsilon$.