# OpenReview forum: "Bicriteria Approximation Algorithms for the Submodular Cover Problem"
_NeurIPS.cc/2023/Conference — NeurIPS 2023 poster_

### Official Review · Reviewer_Swbd · 2023-07-03

**Soundness:** 3 good
**Presentation:** 3 good
**Contribution:** 3 good
**Rating:** 6
**Confidence:** 4

**Summary:**

This work focuses on designing approximation algorithms for the submodular cover problem. In this problem, we are given a submodular function $f$ over subsets of a ground set $U$, and the goal is to find the smallest subset $S$ such that $f(S)$ is greater than or equal to a given threshold. The authors present a fast algorithm for monotone functions with similar approximation guarantees as greedy, while being almost a factor of $n$ faster. They improve the approximation guarantee with an algorithm that runs in exponential time with respect to the size of the optimal solution. They extend their results to non-monotone functions and achieve a high-quality solution with exponential running time.

Furthermore, they consider the regularized submodular cover problem, which is similar to the submodular cover problem minus an additional modular function. They are the first to study this problem and present bicriteria algorithms for this problem. Moreover they compare their algorithms with baselines in multiple experiments.


**Strengths:**

The authors' work makes significant contributions to the field of submodular cover optimization. Their algorithms are simple and easy to understand, and the results are explained clearly. The authors also provide a fair comparison with previous works. The experimental section is well-organized and the datasets used are commonly used in submodular maximization publications in ML conferences. The results show the strengths and weaknesses of this work, and support the theoretical guarantees.


**Weaknesses:**

First, the novelty of the work is not clear to me. The algorithms and proof techniques are similar to previous works and are well-known in the literature. Second, exponential time algorithms are not practical in most cases, and I do not agree with the reasons provided in this work. Third, the results for Regularized SCP are good, but less interesting compared to the rest of the results in this work.


**Questions:**

Please provide details if the weaknesses mentioned above are not accurate.

**Limitations:**

yes

---

> ### Author Rebuttal · Authors · 2023-08-10
>
> Thank you for your time and comments on the paper.
>
> In this paper, we do use a variety of techniques inspired by those found in literature on the cardinality constrained submodular maximization problem (SMP). However, a few examples of components of our algorithms and analyses that are particularly distinct from existing approaches for SMP are: (i) In many algorithms for SMP, the size of the optimal solution is known to be beneath an input budget. This value is used in many algorithms including ones that inspired our algorithms such as the stochastic greedy algorithm of Mirzasoleiman et al. (inspired stoch-greedy-c), and the streaming algorithm of Alaluf et al. (inspired stream-c). In stoch-greedy-c we use a method of adaptively guessing the size of the optimal solution $|OPT|$ throughout the algorithm in order to determine how many samples are needed at each iteration. (ii) Theoretical guarantees for the regularized submodular maximization problem [Harshaw et al.] take an unusual form that is not a typical approximation guarantee. In order to convert these algorithms to ones for the RSCP (convert-reg), we had to use a technique of distorting the function $f$ and then giving it to a regularized submodular maximization subroutine.
>
> We agree that exponential time algorithms are usually impractical. As described in the paper in Section 2.2, it has previously been shown that for general SCP when $f$ is not assumed to be monotone it is not possible for an algorithm to guarantee that its returned solution satisfies $f(X)>\tau/2$ in polynomially many queries of $f$ assuming the value oracle model [Crawford, 2023]. Therefore, in order to have $f(X)$ be nearly feasible, i.e. reach $(1-\epsilon)\tau$, we propose stream-c which requires an exact solution to submodular maximization on an instance of size $O(|OPT|/\epsilon^2)$. When $|OPT|$ is small or in other restricted settings, stream-c may be a reasonable choice.

---

> > ### Comment · Reviewer_Swbd · 2023-08-17
> >
> > I thank the authors for their response.

---

### Official Review · Reviewer_AFAK · 2023-07-06

**Soundness:** 2 fair
**Presentation:** 1 poor
**Contribution:** 2 fair
**Rating:** 4
**Confidence:** 3

**Summary:**

The paper studies bicriteria approximation algorithms for sub-modular cover problem (SCP). In the problem, we are given an oracle to some sub-modular function f, and a threshold tau. Our goal is to find the smallest set X such that f(x) >= tau.

The paper is not well-written at all. There are critical typos in the theorem statements, making it hard to understand what the main results are.  Only after reading the rebuttal and the supplementary materials carefully, I was able to put all the pieces together. Two main results of the paper are

For the monotone SCP problem (MSCP), the result is an algorithm that achieves O(ln(1/eps))-approximation with 1-epsilon violation on the value, using O_eps(n log n) queries. This improves the previous query complexity of O(n^2). However, the algorithm is called threshold-greedy-c, described in Algorithm 5 in the supplementary material. But the theorem statement says the algorithm is threshold-bi, which is never defined.

The second result is for SCP. It says there is an algorithm that achieves 1/epsilon approximation with 1-epsilon violation on the value. The result uses theorem 2, which converts a randomized algorithm for the dual problem SMP into a randomized algorithm for SCP. The algorithm needs to guess a budget g, and repeat the algorithm for SMP multiple times in order to convert the expectation guarantee into high probability guarantee. However, in the theorem statement, gamma should be epsilon as pointed by the authors in the rebuttal, and alpha is never defined. From the supplementary material, I learnt that alpha is the parameter controlling the multiplicative step size of the guessed size budget g, and the term log |OPT| in the running time should really be log_{1 + alpha} |OPT|.

I am raising my score to borderline reject.  There are so many typos in the theorem statements. Yes, they are just a few typos in the whole paper, but I think mistakes in the main theorems, which give the formal descriptions of main results, are intolerable. They are the first things that people will read, and they only take a few lines in the paper.

If I needed to judge the contribution of the results, without taking the typos into consideration, I would say the first result makes a fair improvement, but I do not see too much technical contribution from the second result. It is based on the dual relationship between SCP and SMP, and conversion from expectation to high probability comes from repeating the algorithm multiple times.



**Strengths:**

The paper improved the query complexity for the MSCP algorithm from O(n^2) to O(n log n).

**Weaknesses:**

The paper is not written well.  There are critical typos in the statements of the main theorems.

**Questions:**

No questions.

**Limitations:**

None.

---

> ### Author Rebuttal · Authors · 2023-08-10
>
> We would like to thank the reviewer for their time and comments on the paper. We address each concern and question below.
>
> (1) "The authors stated that they have an algorithm that achieves the best approximation guarantee with O(n ln(n)) queries. I do not see any proof of the result."
>
> We will explain this point more clearly in the next version of the manuscript. It is stated in the paper (see contribution (i) in the introduction) that we provide two algorithms for MSCP that achieve nearly the same theoretical guarantees as the greedy algorithm but in $O(nln(n))$ queries of $f$. The two algorithms that make $O(nln(n))$ queries to $f$ are threshhold-bi and stoch-greedy-c. The theoretical guarantee for threshold-bi is presented in Theorem 1, which states that the number of required queries is $O(\frac{n}{\epsilon}\log(\frac{n}{\epsilon}))$. If we fix $\epsilon$, then the number of queries is $O(n \ln(n))$. The theoretical guarantee for stoch-greedy-c is presented in Theorem 3, which states that the number of required queries is $O\left(\frac{\alpha}{1+\alpha}n\ln(1/\delta)\ln^2(3/\epsilon)\log_{1+\alpha}(|OPT|)\right)$. If we fix $\alpha$, $\delta$, and $\epsilon$ then the number of queries is $O(n \ln(n))$.
>
> (2) "For the problem with (1-epsilon)-approximation allowed on the f(S) value, do you get an O(1/epsilon), or an O(ln(1/epsilon))-approximation guarantee? In result (ii), the size of S is OPT/epsilon. But in Theorem 1, the approximation ratio is ln(2/epsilon) = O(ln(1/epsilon))."
>
> The two approximation ratios that you describe are for different problem settings. If we assume that $f$ is monotone, we get the stronger $O(\ln(n))$ approximation guarantees proven for threshold-bi and stoch-greedy-c (Theorems 1 and 3). If we do not assume that $f$ is monotone and therefore consider the more general setting, then the approximation guarantee we have is the weaker $O(1/\epsilon)$ guarantee proven for stream-c in Theorem 4.  All of the algorithms described above achieves a $(1-\epsilon)$-approximation on the f(S) value (see Theorem 1, Theorem 3 and Theorem 4). There is a typo that may have contributed to the confusion, Theorems 1 and 2 are in the monotone submodular section but say "SC"/"SCP" instead of the correct "MSCP".
>
> (3) "Finally, what is the purpose of Theorem 2? How does it improve the result in Theorem 1? What does gamma mean in the theorem statement?"
>
> The purpose of Theorem 2 is to provide a method for converting randomized algorithms for the dual problem SMP into ones for SCP. In particular, we are interested in using Theorem 2 to convert the stochastic greedy algorithm for monotone SMP [Mirzasoleiman et al.] into an algorithm for MSCP.
>
> We actually made a typo with the $\gamma$ and it should be replaced with $\epsilon$. Thank you for bringing this to our attention. The corrected version of Theorem 2 appears in the supplementary material on page 14 of our submission.
>
> The method described by Theorem 2 gives results beyond those of Theorems 1 and 3 for monotone SCP. In particular, consider the stochastic greedy algorithm of Mirzasoleiman et al., which is a randomized algorithm for SMP that returns a solution set $S$ satisfying $f(S)\geq(1-1/e-t)f(OPT)$ and $|S|=\kappa$. Then in Theorem 2, $\epsilon=2/e+2t$ and $\beta=1$. Then by using the algorithm convert-rand (Algorithm 6 in the appendix) Theorem 2 states that we have a $(1+\alpha, 1-2/e-2t)$-bicriteria that holds with probability $1-\delta$, where $\alpha,\delta$ are input. This is incomparable to the guarantees given in Theorems 1 and 3 since the first part can get arbitrarily close to 1 but the second part, i.e. the guarantee on the feasibility, is relatively weak. In contrast, the approximation ratio on $f$ for both Theorem 1 and Theorem 3 is $1-\epsilon$ and can get arbitrarily close to 1. We will more explicitly describe this example in the next version of the manuscript.

---

> > ### Comment · Reviewer_AFAK · 2023-08-11
> >
> > After reading the paper and supplementary materials more carefully, I am able to put all the pieces together.  Other than the results for regularized SCP and the empirical study, there are two main results given in the paper.
> >
> > The first result (i) is for MSCP. It says there is an algorithm that achieves O(ln(1/eps))-approximation with 1-epsilon violation on the value, using O_eps(n log n) queries. This improves the previous query complexity of O(n^2).  The algorithm is threshold-greedy-c, described in Algorithm 5 in the supplementary material. But the theorem statement says the algorithm is threshold-bi, which is never defined.
> >
> > The second result is for SCP. It says there is an algorithm that achieves 1/epsilon approximation with 1-epsilon violation on the value. The result uses theorem 2, which converts a randomized algorithm for the dual problem SMP into a randomized algorithm for SCP.  The algorithm needs to guess a budget g, and repeat the algorithm for SMP multiple times in order to convert the expectation guarantee into high probability guarantee.  However, in the theorem statement, gamma should be epsilon as pointed by the authors in the rebuttal, and alpha is never defined.  alpha controls the precision of the guessed size budget g, and the running time should have the term log_{1 + alpha} |OPT|.

---

> > > ### Author Response · Authors · 2023-08-11
> > >
> > > We do have a number of different results on several problems, and so we plan on including a table in the next version of the manuscript that makes each contribution and how they fit in more clear. We would like to clarify the main contributions of the paper as follows:
> > >
> > > (1) We have three main results for the monotone submodular cover problem (MSCP) in Section 2.1. These results only apply for MSCP. The first is the algorithm threshold-greedy-c (Algorithm 5 in the appendix) and its guarantees in Theorem 1. The second is the converting method convert-rand that takes randomized algorithms for monotone submodular maximization and converts them into ones for MSCP (Algorithm 6 in the appendix) and its guarantees in Theorem 2. The third is the algorithm stoch-greedy-c (Algorithm 1 in the main text) and its guarantees in Theorem 3. The theorem statements will be edited in order to make it more clear that they apply only to monotone submodular functions (it currently says monotone both in the text and section headings, but some of the theorem statements are unclear). We believe our most interesting result from this section is stoch-greedy-c, which is inspired by the stochastic greedy algorithm for monotone SMP [Mirzasoleiman et al., 2015a] but we use a method of adaptively guessing the size of the optimal solution throughout the algorithm to propose a new sample-efficient algorithm for MSCP.
> > >
> > > (2) We have one main result for the general submodular cover problem (SCP), where $f$ is submodular but not necessarily monotone. This is the algorithm stream-c (Algorithm 2) and its guarantees in Theorem 4.
> > >
> > > (3) Finally, we have a result for the regularized monotone submodular cover problem (RSCP), which does not fall under the setting of SCP since the objective may take on negative values. We propose a method of converting algorithm for the regularized monotone submodular maximization problem, convert-reg (Algorithm 3 in the main text) and its theoretical guarantees are in Theorem 5. We then propose the algorithm distorted-bi for regularized monotone submodular maximization that produces different approximation guarantees compared to existing ones in the literature (see Section 2.3 in the paper for more details on this), in order to be used by convert-reg to produce an algorithm for RSCP.
> > >
> > > In addition, "threshold-bi" is meant to say threshold-greedy-c. We will fix this additional typo in the next version of the manuscript. The alpha in Theorem 2 is an input parameter which we also mentioned in the corresponding pseudocode convert-rand (Algorithm 6 in the supplementary). We will clarify it more clearly in Theorem 2. The reviewer is correct that the term $\log(|OPT|)$ in Theorem 2 should say $\log_{1+\alpha}|OPT|$. We want to point out that most of the typos and ambiguity is coming from Theorem 2 and its corresponding algorithm convert-rand. We believe part of the problem is that much of the information on convert-rand is in the supplementary material. We plan to fix the typos as well as add more discussion about convert-rand and its theoretical guarantees in the main paper.

---

> > > > ### Comment · Reviewer_AFAK · 2023-08-12
> > > >
> > > > I raised my score to weak reject. There are so many typos in the statements of the main theorems, and I think they are intolerable.

---

### Official Review · Reviewer_DEGK · 2023-07-07

**Soundness:** 3 good
**Presentation:** 4 excellent
**Contribution:** 4 excellent
**Rating:** 8
**Confidence:** 3

**Summary:**

This paper proposes several bicriteria approximation methods for solving the standard monotone submodular cover problem (SCP). Additionally, the authors also propose new variants of the problem: removing the monotone assumption, producing nearly feasible solutions, and adding regularization via a modular cost function. The general methodology involves novel conversions from common bicriteria algorithms for the (dual) submodular maximization problem (stochastic greedy, threshold greedy, distorted greedy) into ones for SCP.

**Strengths:**

- Significance: This paper uses a powerful framework which leads to not only novel algorithms but also new problem formulations. Expanding the scope of SMP/SCP conversion is of independent interest and has high potential impact
- Originality/Clarity: Novel combination of previous work and new ideas (discussed and cited clearly)

**Weaknesses:**

- Experiments: The impact of the regularized SCP problem would be more significant if the paper included experimental results comparing convert-reg and distorted-greedy-bi. The paper would be improved if it provided additional motivations/applications for each SCP formulation. When is SCP preferable to SCM?
- Typo: distorted algorithm is distorted-greedy-bi in the main paper and distorted-bi in the appendix

**Questions:**

- What are some applications of regularized SCP?
- Are any bicriteria approximations or query complexity results optimal, or can they be improved further?


-----
EDIT: I have read the author rebuttal, and it addressed my questions sufficiently.

**Limitations:**

No discussions of limitations or broader impact

---

> ### Author Rebuttal · Authors · 2023-08-10
>
> Thank you very much for your time and comments. We address each of your questions below.
>
> - "What are some applications of regularized SCP?"
>
> Many applications of the well-studied regularized submodular maximization problem (SMP) [Harshaw et al., 2019] are also suitable applications for RSCP, just with a shifted emphasis from keeping within a budget (the constraint in regularize SMP) to achieving a certain value of $f$ (the constraint in RSCP). As an example, consider the profit maximization setting where $f=g-c$ and $g$ models the revenue from a group of products while $c$ models the price of producing them. Then RSCP would formalize the problem of achieving a certain amount of revenue ($\tau$) using the minimum amount of products. In addition, many existing applications of SCP such as data summarization [Mirzasoleiman et al., 2016]  or influence in a social network  [Crawford et al., 2019] could be extended to RSCP by adding a modular penalty/cost for including each item into the solution set.
>
> - "Are any bicriteria approximations or query complexity results optimal, or can they be improved further?"
>
> It was proven by [Feige, 1998] that set cover cannot be approximated efficiently below a threshold of $(1-o(1))\ln(n)$ subject to the condition that NP has slightly superpolynomial time algorithms. Set cover is a special case of the general monotone submodular cover problem, and therefore this result applies to MSCP as well as SCP. If we fix the parameter $\epsilon=1/n$, then the algorithms threshold-bi and stoch-greedy-c presented in our paper achieve nearly feasible solutions for MSCP with approximations of $\ln(2n)+1$ (Theorem 1) and $(1+\alpha)\lceil \ln(3n)\rceil$ (Theorem 3) respectively. Therefore these algorithms have bicriteria guarantees that are somewhat close to the best possible. However, we have not yet been able to determine the optimality of our results beyond this.

---

> > ### Comment · Reviewer_DEGK · 2023-08-14
> > **Reply to Rebuttal**
> >
> > The authors have addressed my questions sufficiently.

---

### Official Review · Reviewer_fbw1 · 2023-07-08

**Soundness:** 4 excellent
**Presentation:** 3 good
**Contribution:** 4 excellent
**Rating:** 7
**Confidence:** 2

**Summary:**

The paper considers the Submodular Cover Problem (SCP), where one
is given a submodular function f through an oracle that returns the
value f(A) for each subset A of the underlying set U (over which
the function f is defined). The goal is to find a minimum cardinality
subset X of a set such that the value f(X) is >=  a given threshold
\tau and |X| is a minimum. Since this problem is NP-hard in general,
researchers have considered bicriteria approximation algorithms,
where the size of the size of the returned subset X is within a
small factor of the size of a minimum subset and the value f(X) is
at least a suitable fraction of the given threshold.  While some
such approximation algorithms are available (using a dual problem),
they may use quadratically many queries to the oracle.  The focus
of the paper is to develop methods whose approximation guarantees
are close to known results but which use significantly fewer queries
to the evaluation oracle. Results are provided for general SCP as
well as some restricted versions.

**Strengths:**

(1) The paper presents new methods for SCP with significantly smaller
  numbers of queries to the function evaluation oracle without significantly
  affecting the performance guarantees. To this reviewer's knowledge, this
  is the first work that achieves this goal for SCP. Given the importance
  of submodular optimization in ML these results represent a useful advance.

  (2) The paper nicely summarizes prior work on the topic.

**Weaknesses:**

   A (very) minor weakness is that it takes a fair amount of time
to understand the details regarding the algorithms. (This is due to the
nature of the problem.)

**Questions:**

(1) In the results for the general SCP (lines 50--54, page 2), it is
  mentioned that the algorithm does not run in polynomial time in general.
  Is there a known hardness result which precludes a polynomial
  time algorithm for this case (under some well accepted hypothesis in
  complexity theory)?

  (2) Lines 98--100 in Section 1.2 discusses a result from {Crawford, 2023].
  This result (as described) says that one "cannot have an algorithm for SCP
  which can guarantee that f(X) >= \tau/2 using only polynomially many queries".
  Does this result rely  on an underlying complexity hypothesis (such as P != NP)?

  Some minor suggestions to the author(s):

  (a) In stating the results for RCSP (lines 56--62 on page 2), please
  consider indicating the number of queries to the oracle.

  (b) Line 65: "make a large" ---> "provides a large"

  (c) Line 120 (page 3): "about queries" ---> "with respect to the
      number of queries"

---

> ### Author Rebuttal · Authors · 2023-08-10
>
> Thank you for your time and comments about the paper. We will update the manuscript to add your suggested modifications. We address your questions below.
>
> (1) The result we are aware of for SCP is that under the value oracle model of access for $f$, meaning that $f$ is only accessed as a black box that returns $f(X)$ given $X$, it is not possible for an algorithm to produce a solution $X$ with the guarantee that $f(X)>\tau/2$ in polynomially many queries to $f$ [Crawford, 2023]. Our algorithm stream-c guarantees $f(X)\geq (1-\epsilon)\tau$ for any $\epsilon > 0$, but on the other hand is not guaranteed to make polynomially many queries to $f$.
>
> (2) The result of [Crawford, 2023] does not rely on any complexity hypothesis, but rather applies to any algorithm under the value oracle model. For more intuition about hardness results for submodular optimization problems assuming the value oracle model access to $f$, we refer the reviewer to Section 4 of [Feige et al., 2011]. In fact, the result that we cite from [Crawford, 2023] for general SCP is closely related to Theorem 4.5 in this paper.

---

> > ### Comment · Reviewer_fbw1 · 2023-08-12
> >
> > I have gone through the rebuttal. My questions/concerns have been addressed satisfactorily.

---

### Decision · Program_Chairs · 2023-09-21

**Decision:**

Accept (poster)

**Comment:**

All reviewers appreciate the theoretical improvement of the results for several variants of the important submodular cover problem. The authors are strongly advised to improve the presentation, as suggested by the reviewers.